# Denoising Hamiltonian Network for Physical Reasoning

**Congyue Deng**                                                    *congyue@stanford.edu*
*Stanford University, Massachusetts Institute of Technology*

**Brandon Y. Feng**                                                 *branfeng@mit.edu*
*Massachusetts Institute of Technology*

**Cecilia Garraffo**                                                *cgarraffo@cfa.harvard.edu*
*Harvard-Smithsonian Center for Astrophysics*

**Alan Garbarz**                                                    *alan@df.uba.ar*
*Universidad de Buenos Aires and Instituto de Física de Buenos Aires – CONICET*

**Robin Walters**                                                   *r.walters@northeastern.edu*
*Northeastern University*

**William T. Freeman**                                              *billf@mit.edu*
*Massachusetts Institute of Technology*

**Leonidas Guibas**                                                 *guibas@stanford.edu*
*Stanford University*

**Kaiming He**                                                      *kaiming@mit.edu*
*Massachusetts Institute of Technology*

**Reviewed on OpenReview:** *https://openreview.net/forum?id=KublEgx7Hv*

## Abstract

Machine learning frameworks for physical problems must capture and enforce physical constraints that preserve the structure of dynamical systems. Many existing approaches achieve this by integrating physical operators into neural networks. While these methods offer theoretical guarantees, they face two key limitations: (i) they primarily model local relations between adjacent time steps, overlooking longer-range or higher-level physical interactions, and (ii) they focus on forward simulation while neglecting broader physical reasoning tasks. We propose the Denoising Hamiltonian Network (DHN), a novel framework that generalizes Hamiltonian mechanics operators into more flexible neural operators. DHN captures non-local temporal relationships and mitigates numerical integration errors through a denoising mechanism. DHN also supports multi-system modeling with a global conditioning mechanism. We demonstrate its effectiveness and flexibility across three diverse physical reasoning tasks with distinct inputs and outputs.

## 1 Introduction

Physical reasoning – the ability to infer, predict, and interpret the behavior of dynamic systems – is fundamental to scientific inquiry. Machine learning frameworks designed to address such challenges are often expected to go beyond merely memorizing data distributions, aiming to uphold the laws of physics, account for energy and force relationships, and incorporate structured inductive biases that surpass those of purely data-driven models. Scientific machine learning addresses this challenge by embedding physical constraints directly into neural network architectures, often through explicitly constructed physical operators.

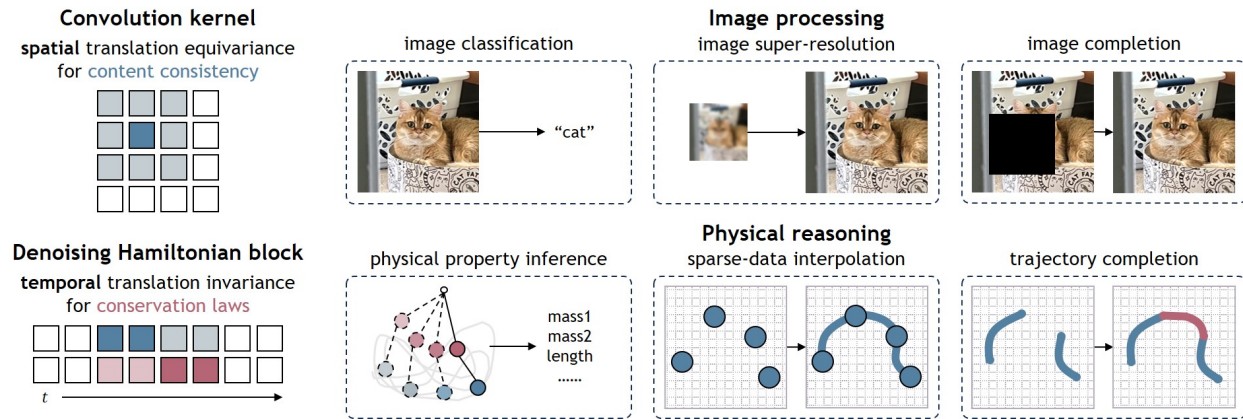

Figure 1: **Denoising Hamiltonian Network (DHN)** generalizes Hamiltonian mechanics into neural operators. It enforces physical constraints while leveraging the flexibility of neural networks, opening pathways for broader applications in physical reasoning.

However, these methods face two key limitations. (i) These methods primarily learn local temporal updates, predicting state transitions from one time step to the next, without capturing long-range dependencies or abstract system-level interactions. (ii) They focus predominantly on forward simulation, forecasting a system's evolution from initial conditions, while largely overlooking complementary tasks such as super-resolution, trajectory inpainting, or parameter estimation from sparse observations. To address these limitations, we aim to design more general neural operators that both follow physical constraints, and unleash the expressivity of neural networks to learn high-level information from data.

To motivate our approach, we draw inspiration from the success of CNNs. Classical image-processing systems utilized handcrafted convolution filters for low-level tasks such as edge detection. The breakthrough came with deep convolutional neural networks (CNNs), which generalized these filters by making them learnable, stacking them with nonlinearities, and expanding them into high-dimensional feature channels. This transition enabled progress from local processing to high-level tasks like object recognition and image generation.

Building on this inspiration, we introduce the **Denoising Hamiltonian Network (DHN)**, a framework that generalizes Hamiltonian mechanics into more flexible neural operators (Fig. 1). Similar to CNNs absorbing parameters from classical image processing (*e.g.*, filter size, stride) into learnable and tunable components, DHN reformulates parameters of numerical integration as architectural hyperparameters, enabling more adaptive and expressive physical modeling. For example, state discretization becomes a learnable block size, and small, fixed time steps are replaced by variable strides, analogous to how CNNs adapt their receptive fields.

More concretely, DHN enforces physical constraints while leveraging the flexibility of neural networks with three key innovations: (i) First, DHN extends Hamiltonian neural operators to capture non-local temporal relationships by treating groups of system states as tokens, allowing it to reason holistically about system dynamics rather than in isolated steps. (ii) Second, DHN introduces a denoising objective analogous to variational integration but learned through data. It mitigates integration errors without relying on additional off-trajectory data for state optimization. Additionally, with different noise patterns, DHN supports flexible training and inference across various task contexts. (iii) Third, we introduce global conditioning to facilitate multi-system modeling, enabling DHN to model heterogeneous physical systems under a unified framework.

To evaluate DHN's versatility, we test it across three distinct reasoning tasks: (i) trajectory prediction and completion, testing long-horizon stability and accuracy; (ii) physical parameter inference from state observations, examining whether the learned representations capture global physical properties; and (iii) trajectory interpolation via progressive super-resolution, assessing the ability to reconstruct physically consistent trajectories from sparse observations. These tasks together also demonstrate that DHN provides a unified framework for diverse physical reasoning problems beyond standard next-state prediction.

In summary, this work moves toward more general network architectures that embed physical constraints beyond local temporal relationships, opening pathways for broader applications in physical reasoning beyond conventional forward simulation and next-state prediction.

## 2 Related Work

Machine learning approaches for physical modeling span from fundamental equations of motion to high-dimensional operator learning. Our work extends Hamiltonian neural networks (HNNs) into a flexible, sequence-based paradigm that enables multi-task inference and generative conditioning.

**Hamiltonian Neural Networks (HNNs)** Scientific machine learning aims to embed physical laws into neural architectures. Hamiltonian Neural Networks (HNNs) (Greydanus et al., 2019) enforce symplectic structure and energy conservation in learned dynamics, inspiring various extensions: Lagrangian Neural Networks (LNNs) (Cranmer et al., 2020), Symplectic ODE-Nets (Zhong et al., 2019), and Dissipative SymODEN (Zhong et al., 2020), which introduce damping terms. Constraints have also been incorporated into HNNs (Finzi et al., 2020b), and some models infer Hamiltonian dynamics directly from image sequences (Toth et al., 2019). Despite their strengths in forward simulation, standard HNNs typically model one system at a time and rely on uniform-step integration, limiting their use in trajectory completion, sparse-data interpolation, or super-resolution.

**Physics-informed and operator-based methods** Beyond Hamiltonian-specific formulations, a broader line of work embeds physical structure directly into neural models, most commonly partial differential equation (PDE) constraints. Another approach embeds partial differential equation (PDE) constraints directly into neural models. Physics-Informed Neural Networks (PINNs) (Raissi et al., 2019) enforce PDE-based losses for solving forward and inverse problems, while Fourier Neural Operators (FNOs) (Li et al., 2020) learn mappings between function spaces using global Fourier transforms. Neural ODEs (Chen et al., 2018; Dupont et al., 2019) parameterize continuous-time dynamics with learnable differential equations. As the term "physics-informed neural network" has expanded to cover a wide range of models and applications, much of the literature has focused on fitting a single system or solving a specific PDE instance Bonev et al. (2023), with emphasis on the fitting accuracy. In contrast, our work targets a different regime of physical learning: given training data from many systems of similar form but varying parameters, can a model form high-level understandings that can generalize to previously unseen systems while still preserving meaningful physical guarantees?

**Symmetry-constrained neural networks** From a broader perspective, the conservation laws enforced in Hamiltonian-based models can be interpreted as symmetry constraints, grounded in Noether's theorem Noether (1971), which establishes a direct connection between continuous symmetries and conserved quantities Greydanus et al. (2019); Cranmer et al. (2020). This viewpoint places Hamiltonian neural networks within the more general framework of equivariant learning, where network architectures are designed to respect prescribed group actions on inputs and outputs Cohen & Welling (2016); Kondor & Trivedi (2018). Historically, convolutional neural networks provided one of the earliest and most successful examples of such guarantees by enforcing translation equivariance, a principle that has since been extended beyond Euclidean domains to graphs, manifolds, and other structured data types LeCun et al. (2002); Bronstein et al. (2017).

More recently, research on equivariant architectures has expanded to a wider range of symmetry groups Cohen et al. (2019); Weiler et al. (2021); Deng et al. (2021); Thomas et al. (2018); Geiger & Smidt (2022), and importantly, has begun to explore relaxations of exact equivariance, allowing models to balance inductive bias with expressive flexibility in settings where symmetries may be approximate, partial, or system-dependent Finzi et al. (2020a); Romero & Lohit (2022); Kaba & Ravanbakhsh (2023); Pertigkiozoglou et al. (2024). Our work aligns with this latter direction: rather than enforcing strict equivariance with respect to predefined conservation laws, DHN integrates physically motivated constraints while retaining the capacity to generalize across physical systems.

**System identification and multi-system modeling** Learning from heterogeneous physical systems requires system identification, traditionally performed via parametric models (Ljung, 1999) or hybrid PDE-

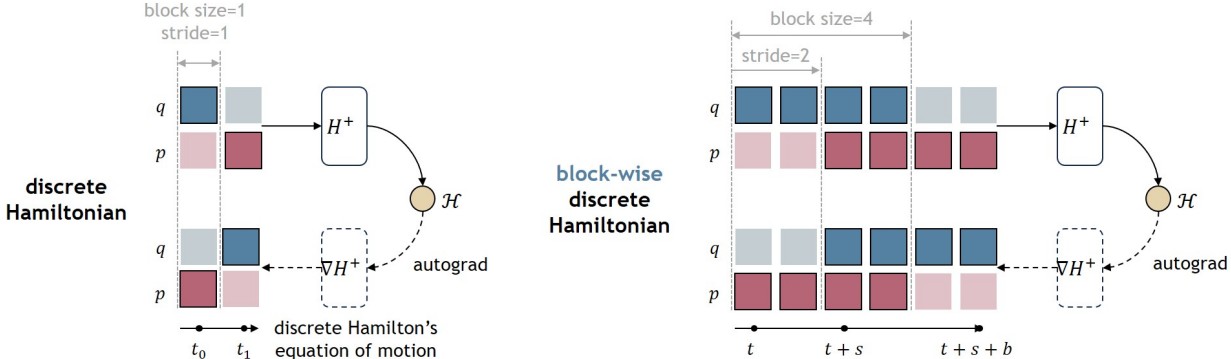

Figure 2: **Block-wise Hamiltonian.** Left: Classical HNN with the discrete Hamiltonian, which can be viewed as a special case with block size $b = 1$ and stride $s = 1$. Right: A discrete (right) Hamiltonian block with $b = 4, s = 2$. Dark blue and dark red indicate network inputs and outputs. Light colors are for placeholder states.

constrained approaches (Raissi et al., 2019). While Hamiltonian methods implicitly encode system parameters through energy landscapes, conventional HNNs often require training separate models per system. We introduce a generative conditioning mechanism via a learned latent code, enabling a single model to generalize across multiple systems while preserving inductive biases from Hamiltonian dynamics.

## 3 Method

We aim to design neural operators that respect physical constraints while leveraging the flexibility and expressivity of neural networks. To this end, we extend Hamiltonian operators to a block-wise form that learns more flexible state relations with generalized conserved quantities (Sec. 3.2). We then introduce a denoising mechanism that replaces the state optimization in variational integration with a learned process; using different masking patterns, it also enables adaptive inference strategies across tasks within a unified framework (Sec. 3.3). Finally, we incorporate global conditioning to extract high-level information from physical trajectories (Sec. 3.4).

### 3.1 Preliminaries

**Learning with Hamiltonian mechanics**  Let's start with *phase-space coordinates* $(q, p)$, where $q$ is the *generalized coordinates* and $p$ is the *generalized momenta* or *conjugate momenta*. If $q$ represents the particle positions in Euclidean coordinates, then $p$ corresponds to their linear momenta. If $q$ represents angular positions in spherical coordinates, $p$ corresponds to the associated angular momenta. We consider the time-invariant *Hamiltonian*, which is a scalar function $\mathcal{H}(q, p)$ satisfying

$$\frac{\mathrm{d}q}{\mathrm{d}t} = \nabla_p \mathcal{H}, \quad \frac{\mathrm{d}p}{\mathrm{d}t} = -\nabla_q \mathcal{H}. \tag{1}$$

Eq. 1 is known as Hamilton's equations of motion and describes system evolution by defining a trajectory in phase space along the vector field $(\nabla_p \mathcal{H}, -\nabla_q \mathcal{H})$. This field, called the *symplectic gradient*, governs the dynamics such that movement along $\mathcal{H}$ induces the most rapid change in the Hamiltonian, whereas motion in the symplectic direction preserves the system's energy structure.

Hamiltonian Neural Networks (HNN) Greydanus et al. (2019) treat the Hamiltonian as a black-box function $\mathcal{H}(q, p; \theta)$ parameterized by a neural network and optimize the network parameters to minimize the loss function

$$\mathcal{L}_{\mathrm{HNN}}(\theta) = \left\| \nabla_p \mathcal{H} - \frac{\mathrm{d}q}{\mathrm{d}t} \right\| + \left\| \nabla_q \mathcal{H} + \frac{\mathrm{d}p}{\mathrm{d}t} \right\|. \tag{2}$$

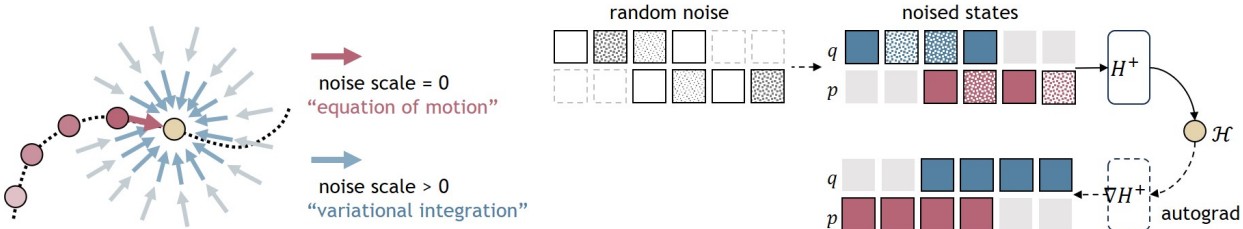

Figure 3: **Denoising Hamiltonian block.** The model integrates the Hamiltonian equation of motion (pink arrow) and state optimization (blue arrow) into a unified framework (left) by learning state denoising conditioned on different noise scales (right).

Starting with an initial state $(q_0, p_0)$, one can compute the trajectory $(q_t, p_t)$ by integrating the symplectic gradient $(\nabla_p \mathcal{H}(q_t, p_t; \theta), -\nabla_q \mathcal{H}(q_t, p_t; \theta))$ over time $t$.

**Discrete Hamiltonian**  Aside from the continuous Hamiltonian $\mathcal{H}$ and its discretizations, one can also directly define the discrete Hamiltonian with discrete mechanics and duality theory in convex optimization Gonzalez (1996). The discrete "right" Hamiltonian $H^+$ gives the equation of motion in the form

$$q_{t+1} = \nabla_p H^+(q_t, p_{t+1}), \quad p_t = \nabla_q H^+(q_t, p_{t+1}). \tag{3}$$

The "right" means that $q$ is forward and $p$ is backward in time. This formulation serves as a first-order discrete approximation of the continuous Hamiltonian $\mathcal{H}$ by

$$q_{t+1} = q_t + \Delta t \nabla_p \mathcal{H}(q_t, p_{t+1}), \quad p_t = p_{t+1} + \Delta t \nabla_q \mathcal{H}(q_t, p_{t+1}). \tag{4}$$

Fig. 2 left illustrates a discrete right Hamiltonian network for computing the state relations between time steps $t_0$ and $t_1$. In the following sections, we describe our network design primarily using the right Hamiltonian $H^+$, but similar equations can also define the left Hamiltonian $H^-$. Additional details can be found in the Appendix.

**Our motivations**  Exemplified by HNN, physical networks generally learn the state relations between adjacent time steps $t$ and $t + 1$ modeled by an update rule $(q_{t+1}, p_{t+1}) = \texttt{update\_rule}(q_t, p_t)$. Compared to forward modeling, the discretization in Eq. 3 is more accurate and better preserves the symplectic structure of the system under temporal integrations. However, the implicit nature of these update rules poses inference-time challenges, as determining new system states requires solving an optimization problem, which is difficult only with data points on the simulation trajectory but without additional reference points outside the trajectory. Our solution is to incorporate the optimization process into the network, leading to the *denoising* Hamiltonian network (Sec. 3.3) that unifies the state optimization at each time step and the Hamiltonian-modeled state relations across time steps.

### 3.2 Block-wise Discrete Hamiltonian

We define state blocks as a stack of $(q, p)$ states concatenated along the time dimension $Q_t^{t+b} = [q_t, \cdots, q_{t+b}]$, $P_t^{t+b} = [p_t, \cdots, p_{t+b}]$, with $b$ being the block size. We also introduce the stride $s$ as a hyperparameter that can be flexibly defined, replacing the fixed time interval $\Delta t$ in Eq. 4. This approach enables the network to capture broader temporal correlations while preserving the underlying Hamiltonian structure. We define our block-wise discrete (right) Hamiltonian by relating two overlapping blocks of system states, each of size $b$, with a shift stride of $s$

$$Q_{t+s}^{t+s+b} = \nabla_P H^+(Q_t^{t+b}, P_{t+s}^{t+s+b}), \quad P_t^{t+b} = \nabla_Q H^+(Q_t^{t+b}, P_{t+s}^{t+s+b}). \tag{5}$$

Fig. 2 illustrates a block-wise discrete Hamiltonian of a block size $b = 4$ and a stride $s = 2$. Classical HNNs can be viewed as a special case of block size $b = 1$ and stride $s = 1$. Physical interpretations of the block-wise

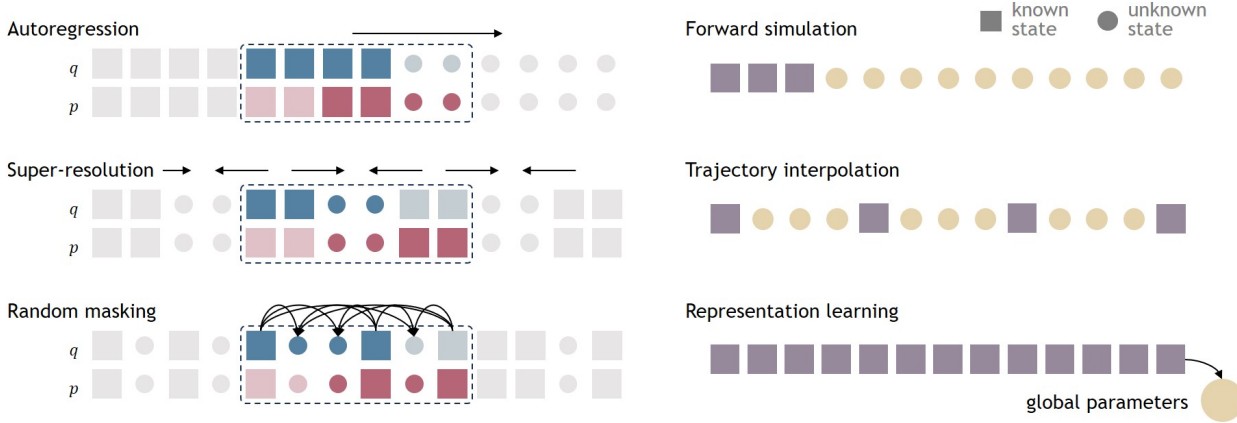

Figure 4: **Different masking patterns (left) adapt to different reasoning tasks (right).** Blocks with pinks and blues surrounded by dotted lines are the denoising Hamiltonian blocks.

Hamiltonian with $b > 1, s > 1$ and theoretical derivations of its conserved quantities can be found in the Appendix. Similar to the equation-motion loss for HNN (Eq. 5), a block-wise discrete Hamiltonian network $H_\theta^+$ can be trained with the equation-of-motion loss

$$\mathcal{L}_{\mathrm{eom}}(\theta) = \left\| \nabla_P H_\theta^+(Q_t^{t+b}, P_{t+s}^{t+s+b}) - Q_{t+s}^{t+s+b} \right\| + \left\| \nabla_Q H_\theta^+(Q_t^{t+b}, P_{t+s}^{t+s+b}) - P_t^{t+b} \right\|. \tag{6}$$

### 3.3 Denoising Hamiltonian Network

**Masked modeling and denoising** Following our motivations introduced in Sec. 3.1, we want the Hamiltonian blocks to not only model the state relations across time steps, but also learn the state optimization per time step for inference. To achieve that, we adopt a masked modeling strategy He et al. (2022) by training the network with a part of the input states masked out (Fig. 3).

Concretely, taking the blocked input state $Q_t^{t+b}$ as an example, we define known and unknown states in this block with a binary mask $M_t^{t+b} = [m_t, \cdots, m_{t+b}]$, with 0 for unknown states and 1 for known states. Rather than simply masking out the unknown states, we perturb them with noise sampled at varying magnitudes and learn to denoising them with iterative refinements (Fig. 3 left). With a sequence of increasing noise levels $0 = \alpha_0 < \alpha_1 < \cdots < \alpha_N = 1$, we randomly sample Gaussian noises $\mathcal{E}_t^{t+b} = [\varepsilon_t, \cdots, \varepsilon_{t+b}]$ and noise scales $A_t^{t+b} = [\alpha_t, \cdots, \alpha_{t+b}]$. Applying the noise to the unknown states, we get the final perturbed state $\widetilde{Q}_t^{t+b}$ with

$$\tilde{q}_s = \begin{cases} q_s & \text{if } m_s = 1 \\ (1 - \alpha_s)q_s + \alpha_s \varepsilon_s & \text{if } m_s = 0 \end{cases} \quad (\text{for } s = t, \cdots, t + b). \tag{7}$$

Similarly, we can get the perturbed state $\widetilde{P}_t^{t+b}$, and together with $\widetilde{Q}_t^{t+b}$ define the denoising loss

$$\mathcal{L}_{\mathrm{denoise}}(\theta) = \left\| \nabla_P H_\theta^+(\widetilde{Q}_t^{t+b}, \widetilde{P}_{t+s}^{t+s+b}) - Q_{t+s}^{t+s+b} \right\| + \left\| \nabla_Q H_\theta^+(\widetilde{Q}_t^{t+b}, \widetilde{P}_{t+s}^{t+s+b}) - P_t^{t+b} \right\|. \tag{8}$$

This encourages the network $H_\theta^+$ to learn a conditioned state optimization, with the known states with mask values $m_s = 1$ as conditions and the unknown ones with $m_s = 0$ as the states to denoise (Fig. 3 right). Combining Eq. 6 and Eq. 8, the final loss function for the network is $\mathcal{L}_{\mathrm{eom}}(\theta) + \mathcal{L}_{\mathrm{denoise}}(\theta)$.

To note, unlike the denoising diffusion model that learns multi-modal distributions, this denoising process has a fixed target and thus doesn't require many denoising steps (we set it to be 10 in our experiments). More details for training and inference are in the Appendix.

**Different masking patterns** By designing distinct masking patterns during training, we enable flexible inference strategies tailored to different tasks. Fig. 4 shows three types of different masking patterns:

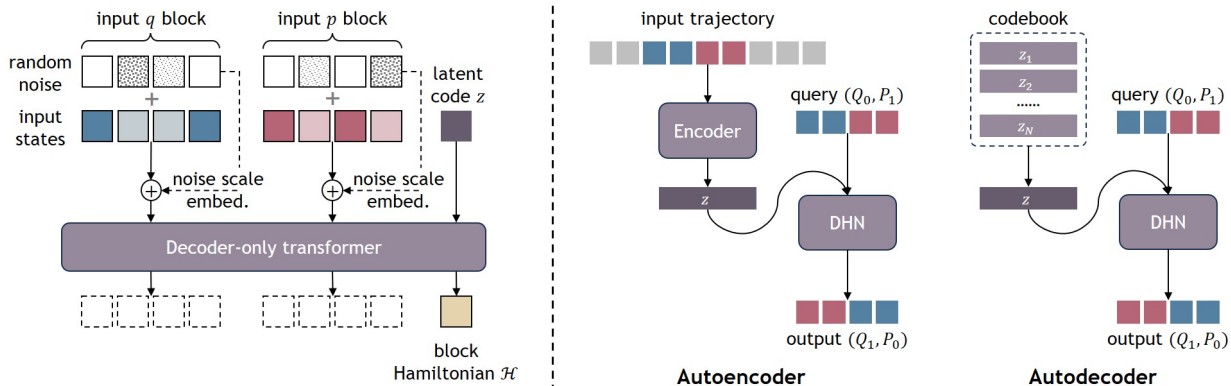

Figure 5: **Network architecture. Left: Decoder-only transformer.** We use a latent code $z$ for each trajectory to serve as the query token for the Hamiltonian value output. Per-state noise scales are encoded and added to the positional embeddings. Dark purples (in all shades) indicate trainable modules or variables. **Right: Autodecoder.** Instead of encoding the input trajectory with an encoder, we maintain a codebook for the entire dataset with a learnable latent code for each trajectory. Dark purples (in all shades) indicate trainable modules or variables.

*autoregression* by masking out the last few states of a block, which resembles physical simulation in terms of next-state prediction with forward modeling; *super-resolution* by masking out the states in the middle of a block, which can be applied to data interpolation; and more generally, *arbitrary-order* masking including random masking, which can adapt to the different tasks.

### 3.4   Network Architecture

**Decoder-only transformer**   For each Hamiltonian block, the network inputs are a stack of $Q_t^{t+b}$ of different time steps, a stack of $P_{t'}^{t'+b}$, and we also introduce a global latent code $z$ for the entire trajectory as conditioning. We employ a decoder-only transformer Radford et al. (2019); Jin et al. (2024), which resembles a GPT-like decoder-only architecture but without a causal attention mask, as shown in Fig. 5 left. We apply self-attention to all input tokens $[Q_t^{t+b}, P_{t'}^{t'+b}, z]$ as a sequence of length $2b + 1$. Analogous to the CLS token in Vision Transformers (ViT) Dosovitskiy (2020), we employ a global latent code $z$ to serves as the query token for outputting the Hamiltonian value $\mathcal{H}$. We also encode the per-state noise scales into the network by adding their embeddings to the positional embedding.

**Autodecoding**   Rather than relying on an encoder network to infer the global latent code from the trajectory data, we adopt an autodecoder framework Park et al. (2019), maintaining a learnable latent code $z$ for each trajectory (Fig. 5 right). This approach allows the model to store and refine system-specific embeddings efficiently without requiring a separate encoding process. During training, we jointly optimize the network weights and the codebook. After training, given a novel trajectory, we freeze the network weights and only optimize the latent code for the new trajectory.

## 4   Experiments

We test our model with three different tasks corresponding to the three different masking patterns, as illustrated in Fig. 4. They are (i) next-state prediction (autoregression) for forward simulation (Sec. 4.1), (ii) representation learning with random masking for physical parameter inference (Sec. 4.2), and (iii) progressive super-resolution for trajectory interpolation (Sec. 4.3). These tasks highlight DHN's adaptability to diverse physical reasoning challenges, testing its ability to generate, infer, and interpolate system dynamics under varying observational constraints. More concretely, we will demonstrate the following:

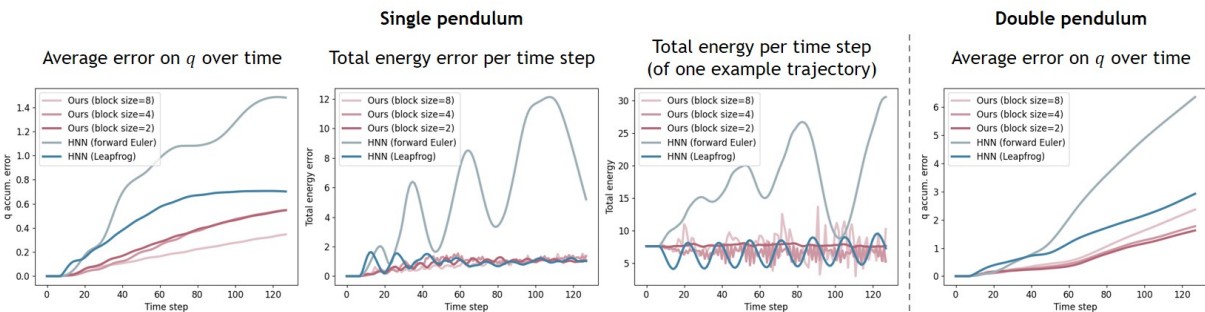

Figure 6: **Forward modeling: fitting known trajectories.** The results of our method are shown in pink, and the results of HNN with different numerical integrators are shown in different shades of blue. The 2nd column shows the error of total energy for the single pendulum system, calculated with state $(q_t, p_t)$ at each time step analytically. The 3rd column shows the total energy predicted by the network over time steps (which wasn't supervised during training) on one example trajectory.

- *Task flexibility.* A single DHN framework supports a range of physical reasoning tasks (Sec. 4.1–4.3). Taken together, these experiments illustrate the breadth of capabilities enabled by our formulation, beyond performance on any single task in isolation.
- *Physically meaningful latent representations.* DHN learns latent representations that capture higher-level physical properties, rather than only low-level local relations between nearby states (Sec. 4.2).
- *Balanced inductive bias and expressivity.* Across all tasks, DHN achieves improved performance by balancing explicit physical constraints with the flexibility of black-box operator designs.

**Setups** We evaluate our model with two settings: the *single pendulum* and the *double pendulum.* Both settings comprise a dataset of simulated trajectories. The single pendulum is a periodic system where the total energy at each state can be directly computed from $(q, p)$, and thus we use it to evaluate the models' energy conservation ability. The double pendulum is a chaotic system where small perturbations can lead to diverged future states. More detailed experimental setups are in the Appendix.

Unlike prior works Toth et al. (2019) that generate data using a fixed set of system parameters while varying initial conditions, we introduce variation by altering the string lengths of the pendulums while keeping initial states fixed. This modification evaluates whether models can generalize to a broader class of parameterized dynamical systems rather than fit into a single-instance system. For both settings, we split the dataset into 1000 training trajectories and 200 testing trajectories. Each trajectory is discretized into 128 time steps. More details can be found in the Appendix.

### 4.1 Forward Simulation

We start with the forward simulation task, where the model predicts the future states of a physical system step-by-step given the initial conditions. We implement this by applying a masking strategy within each DHN block, where the last few tokens are masked during training, requiring the model to iteratively refine and denoise them (Fig. 4 top). For one DHN block of block size $b$ and stride $s$, the mask is applied to the last $b - s$ tokens. At inference time, given the known states at time steps $[0, \cdots, t]$, we apply the DHN block to the time steps $[t - b + 1, \cdots, t + s]$, where we use the known states $[t - b + 1, \cdots, t]$ to predict the unknown states $[t + 1, \cdots, t + s]$. We experiment with block sizes $b = 2, 4, 8$ with strides $s = b/2$.

**Fitting known trajectories** We first evaluate the model's capability to represent known physical trajectories with forward modeling. In this experiment, we train the model to fit 1000 training trajectories, and we test by giving the first 8 time steps of each trajectory and using the model to predict the future 120 steps. As all models are only trained with states of nearby time steps (pairs of adjacent time steps for the baselines, and blocks of $b + s$ states for DHN), small fitting errors can accumulate over time in forward modeling. Beyond

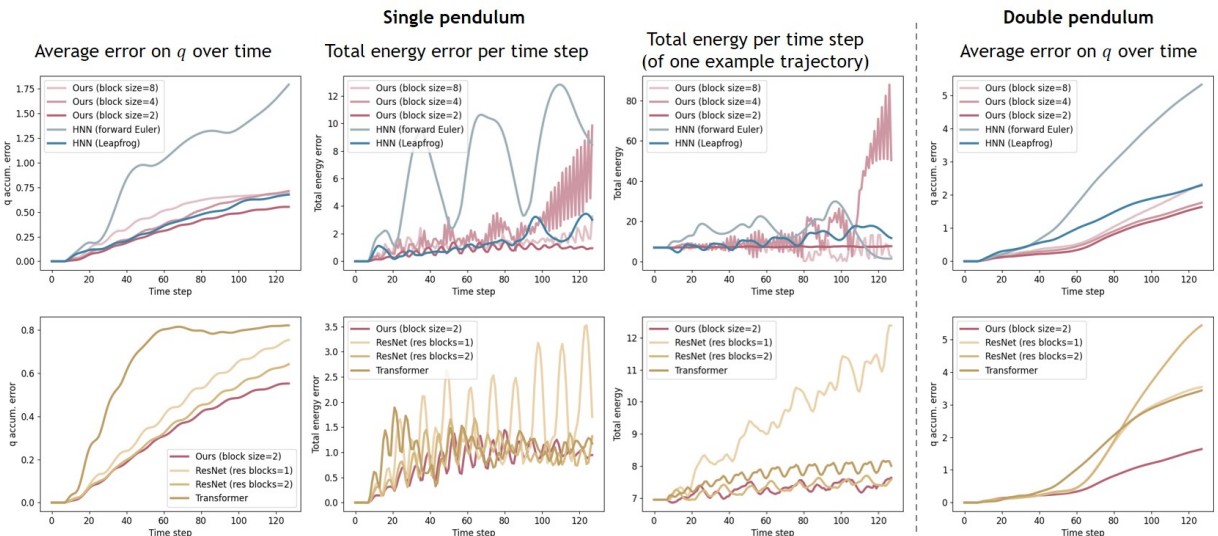

Figure 7: **Forward modeling: completion on novel trajectories.** Top row: Comparison between our method (shown in pink) and HNN with different numerical integrators (shown in blue). Bottom row: Comparison between our method (shown in pink) and vanilla networks with different architectures (shown in yellow). The vanilla networks directly predict the next state $(q_{t+1}, p_{t+1})$ from the current state $(q_t, p_t)$ with one feedforward step. Note that the y-axis scales between the two rows are different.

accumulated prediction errors inherent to the network, inaccuracies also arise from numerical integration approximations, which can amplify deviations over time.

Fig. 6 shows the results of our model with different block sizes, compared to HNN Toth et al. (2019) with different numerical integrators. Left and right are the mean squared error (MSE) on the $q$ predictions at each time step for the single and double pendulum systems, respectively. The middle plots show the averaged total energy error and the evolution of total energy on one example trajectory. Although HNN is a symplectic network with guaranteed energy conservation, the numerical integrator can still induce uncontrollable energy drifts. This additional numerical error is inevitable with forward methods. As mentioned in Sec. 3.1, while this can be addressed by variational integration methods, such implicit state optimizations require off-trajectory data, whereas DHN achieves similar effects through a denoising mechanism without this overhead. With block size $b = 2$, our model conserves the total energy stably. Increased block sizes can cause energy fluctuations at long time ranges, but this fluctuation doesn't show an obvious inclination of energy drift.

**Completion on novel trajectories** We then evaluate our models on novel trajectories with partial observations. Concretely, we give the first 16 time steps in each testing trajectory and use them to optimize for the per-trajectory global latent codes with the network weights frozen, as described in Sec. 3.4. After optimizing these latent codes, we use them to predict the next 112 time steps. This task evaluates DHN's ability to infer system dynamics from sparse initial observations and accurately forecast future states.

Fig. 7 shows our results compared to HNN (top row) and various baseline models without physical constraints (bottom row). Our DHN with small block sizes shows more accurate state prediction with better energy conservation compared to both baselines. Large block sizes can cause error explosion at long time ranges as it is hard for our simple 2-layer network to fit very complex multi-state relations.

**Higher-dimensional system** Figure 8 left illustrates the experimental setup. We consider a one-dimensional spring chain with 10 particles, where a transverse wave propagates from right to left. All particles share the same mass except for one "weird" particle, whose index and mass are randomly sampled. This design produces a family of related systems with localized variations in physical properties. DHN is evaluated on the forward simulation task, using the same training and testing protocol as in the previous experiment in this section (completion on novel trajectories).

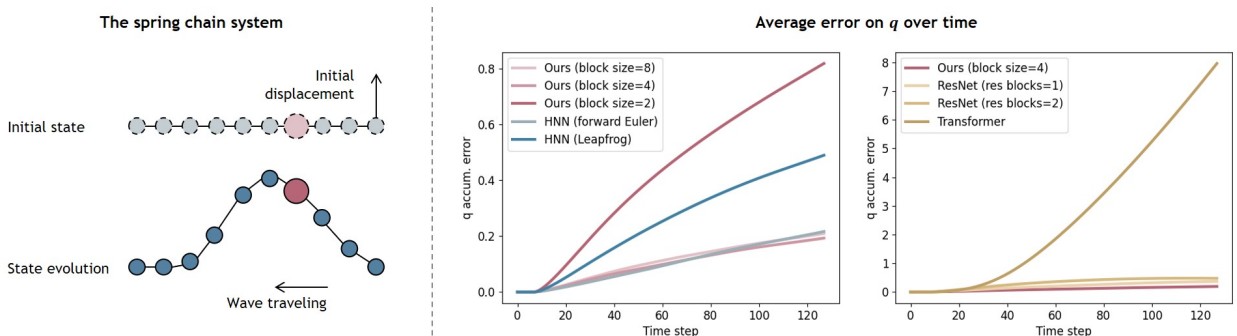

Figure 8: **Traveling wave on spring chains. Left:** Illustration of the system. Blue dots are normal particles of identical mass, and the pink dot is the "weird particle", whose index and mass are randomly sampled. **Right:** Averaged $q$ prediction errors in MSE.

Figure 8 right reports the average mean squared error of positions $q$ over time. DHN with larger block sizes (4 or 8) achieves the lowest error, particularly at longer time horizons. Predicting multiple steps in parallel reduces the number of autoregressive rollouts and consequently mitigates temporal error accumulation. Notably, HNN with Leapfrog integration performs worse than with forward Euler, despite Leapfrog being a higher-order integrator. Since the test set consists of previously unseen physical systems, this indicates that the primary performance bottleneck arises from model generalization across systems rather than numerical integration accuracy. In addition, DHN consistently outperforms unconstrained neural baselines, highlighting the benefit of incorporating physical structure.

The 10-particle spring chain is chosen deliberately for its simple and nearly linear spatial structure, which can be effectively modeled by an MLP. In our architecture, the transformer primarily captures temporal relationships, while spatial interactions are handled by a lightweight network. Although DHN is not inherently limited by system dimensionality, practical performance depends on how well spatial structure is represented. For more complex multi-particle or high-dimensional systems, stronger spatial inductive biases are likely beneficial, for example, permutation-invariant modules such as PointNet, graph neural networks for arbitrary connectivity, and fully spatiotemporal transformers that jointly model spatial and temporal interactions. These directions offer a natural path toward scaling DHN to more general physical systems.

## 4.2 Representation Learning

Next, we test the model's ability to effectively encode and distinguish the parameters of different physical systems. Denoising and random masking are well-established techniques in self-supervised learning, producing state-of-the-art representations in language modeling Devlin (2018) and vision Vincent et al. (2008); He et al. (2022). Here, we apply the random masking pattern (Fig. 4 bottom) and study whether similar paradigms can enhance representation learning in dynamic physical systems.

To quantify the quality of the learned representations, we follow the widely adopted self-supervised representation learning paradigm in computer vision Chen et al. (2020); Oord et al. (2018); He et al. (2020); Kolesnikov et al. (2019) with feature pre-training and linear probing. Specifically, we pre-train the autodecoder alongside the codebook using the training set, then freeze the learned representations and train a simple linear regression layer on top to predict system parameters. This approach assesses whether DHN's latent codes capture meaningful physical properties. We experiment with the double pendulum and predict the length ratio $l_2/l_1$, because this physical quantity is dimensionless and therefore invariant under normalizations in data preprocessing.

Fig. 9 left shows the linear probing results of our DHN with different block sizes (with $s = b/2$), compared to the HNN and vanilla networks. Our model achieves a much lower MSE compared to the baseline networks. As illustrated in Fig. 2, HNN can be viewed as a special case of our Hamiltonian block with kernel size and stride being 1, which is of the most locality. The block sizes and strides we introduce allow the model to

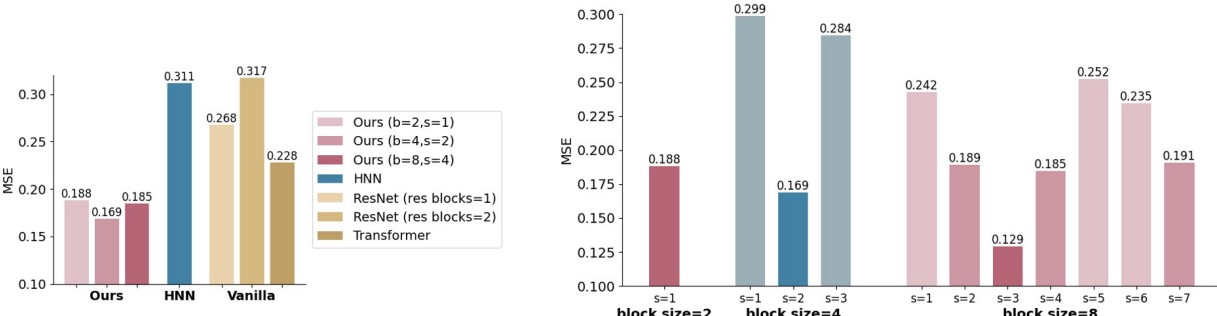

Figure 9: **Linear probing on latent codes (MSE ↓).** We predict $l_2/l_1$ by applying a linear regression layer to the global latent code. **Left:** Comparison with baselines. **Right:** DHN results with different block sizes and strides. Better results are achieved with block size $b$ and stride $s$ around $s \approx b/2$.

observe the system at different scales. In this double pendulum system, a block size of 4 is the best temporal scale for inferring its parameters.

Fig. 9 right shows DHN results with varying block sizes and strides. An explanation is in the "overlap" between the input and output states of $b - s$ time steps. The generalized energy conservation in DHN relies on overlapping regions having identical inputs and outputs, enforced during training via the state prediction loss. Larger overlaps strengthen self-coherence but reduce focus on inter-state relations, while smaller overlaps (with larger strides) encourage learning from distant states but weaken self-coherence, affecting stability. At the extreme, full overlap with zero stride reduces training to enforcing self-coherence alone with identical inputs and outputs. HNN is the special case at another extreme with zero overlap. The results suggest that the optimal block sizes and strides are around $s \approx b/2$, leading to moderate overlaps.

## 4.3 Trajectory Interpolation

To demonstrate the flexibility of the DHN block, we show trajectory interpolation (super-resolution) with the masking pattern in Fig. 4 middle. Learning-based super-resolution is useful when naive interpolation methods (*e.g.*, linear or quadratic) are insufficient to recover the underlying structure of the data. In computer vision, super-resolution methods go beyond simple pixel interpolation to recover semantic content and fine-grained details from low-resolution images (*e.g.*, Ledig et al. (2017); Lim et al. (2017)). In this work, we consider an analogous setting for physical trajectories: given sparsely observed temporal states, we aim to reconstruct high-resolution, physically meaningful trajectories rather than merely filling in missing values numerically.

We conduct $4\times$ super-resolution by repeatedly applying $2\times$ super-resolutions, as shown in Fig. 10 left. Each trajectory is associated with a global latent code, shared across all three stages. During training, both the network weights and these latent codes are optimized jointly with all stages (0, 1, 2). At inference time, given a novel trajectory with known states only at the sparsest level (stage 0), we freeze all network weights in the DHN blocks and optimize for the global latent code with stage 0. After this test-time optimization (autodecoding), we apply the DHN blocks of stages (1, 2) to predict the unknown states in between the known states.

We evaluate the models with two test settings: (i) trajectories with the same initial states as the training ones, and (ii) trajectories of unseen initial states. To set this up, we crop all training trajectories to time steps $[0, \cdots, 64]$. For each trajectory in the test set, we divide it into two segments: time steps $[0, \cdots, 64]$ and $[65, \cdots, 128]$, the former having the same initial state as the training set and the latter having different initial states.

We compare our model to a Convolutional Neural Network (CNN) for super-resolution. Fig. 10 shows our results. For the trajectories with the same initial state as the training data, both models show good interpolation results with lower MSEs. The baseline CNN shows slightly better results, as it has no regularization in itself and can easily overfit the training trajectories. For testing trajectories with unseen

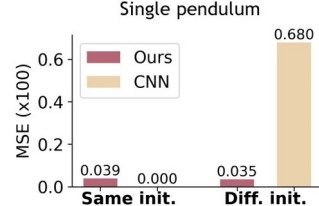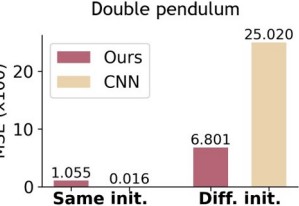

Figure 10: **Interpolation via progressive super-resolution (MSE↓).** Left: The three stages for $2\times$ super-resolution repeated twice. Right: Result comparison between DHN (*Ours*) and a CNN-based super-resolution network (*CNN*). Results are MSE multiplied by 100.

initial states, the CNN struggles to generalize, as its interpolations rely heavily on the training distribution. In contrast, DHN demonstrates strong generalization, as its physically constrained representations enable it to infer plausible intermediate states even under distribution shifts.

## 5 Discussions and Conclusions

Balancing flexibility with physical constraints is key to advancing physics-based learning. Inspired by unified architectures in vision and NLP (*e.g.*, CNNs, transformers), we ask: can a single model handle tasks ranging from global parameter inference to local state prediction, while preserving physical consistency? This motivates three broader questions beyond DHN:

- *What defines physical reasoning in deep learning?* Beyond next-state prediction, it encompasses parameter estimation, system identification, and discovering high-level relationships in dynamical systems. We aim for unified physical learning frameworks that adapt to diverse tasks while enforcing fundamental physical principles.
- *What is physical simulation?* Simulation doesn't always require generating trajectories step by step from an initial state. Even if used sequentially, a full trajectory can be generated in any order. For example, video generation with denoising models produces all frames at once, and Diffusion Policy Chi et al. (2023) predicts chunks of future actions ahead of time, even though they're executed step by step.
- *What physical attributes should a neural network possess?* While PDEs describe systems through local temporal relations, directly using them in networks imposes the same locality. But networks aren't limited to this. Instead of strictly following physical operators, we can extend them, preserving core properties while relaxing others, to design more flexible architectures.

We believe that physics-based learning is on the verge of a major transformation, similar to the rise of self-supervised learning in vision and NLP. By reframing physical reasoning as a reconstruction problem, predicting system states from partial or corrupted inputs, we move toward a unified framework that blends deep learning flexibility with the rigor of physical laws. We also provide more discussions in the Appendix.

**Limitations and future work**   While our current work provides increased flexibility in Hamiltonian-based network designs, it incurs a higher computational cost due to intensive gradient evaluations inherited from HNN. In addition, current experiments focus on small-scale systems with simple temporal dynamics. Scaling to complex spatial-temporal systems may benefit from hierarchical or attention-based architectures inspired by modern vision models.

### Broader Impact Statement

This work aims to advance scientific studies by developing AI tools for physics-based reasoning. By incorporating physical constraints into neural networks, we seek to improve the interpretability and reliability of learning-based models in scientific contexts. However, as with other machine learning approaches, applying neural networks to scientific problems requires caution, as they can produce hallucinations that potentially lead to incorrect conclusions if not properly validated. While incorporating physical constraints can mitigate

such risks, it does not eliminate the need for rigorous verification. It remains essential that results from AI-assisted analyses are evaluated critically and supported by both physical principles and empirical evidence.

**Acknowledgments**

We thank Rell the cat for her photo in Figure 1. We also thank Tianwei Yin, Tianyuan Zhang, Shivam Duggal, Yichen Li, Carolina Cuesta-Lázaro, and Katherine L. Bouman for their helpful discussions. C. Deng and L. Guibas are in part supported by the Toyota Research Institute University 2.0 Program and a Vannevar Bush Faculty Fellowship. B. Y. Feng and W. T. Freeman are in part supported by the NSF Award 2019786 (The NSF AI Institute for Artificial Intelligence and Fundamental Interactions) and the NSF CIF Award 1955864 (Occlusion and Directional Resolution in Computational Imaging). C. Garraffo is funded by AstroAI at the Center for Astrophysics at Harvard & Smithsonian. A. Garbarz is supported by UBA and CONICET and through the grants PICT 2021-00644, PIP 112202101 00685CO, and UBACYT 20020220400140BA. R. Walters is supported by NSF 2134178.

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

# A  Additional Discussions and Motivations

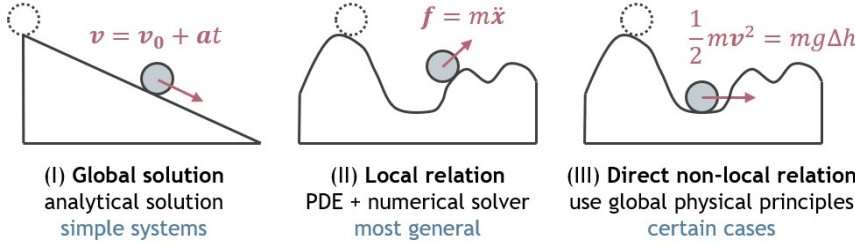

Figure 11: **How can we solve for a physical state?** (I) Closed-form analytical solutions for simple systems. (II) For more complex physical systems, most physical PDEs only model local relations of close-by time steps. (III) For certain physical systems, states can be directly related even if they are not close in time.

Our goal is to design more general neural operators that both follow physical constraints and unleash the flexibility and expressivity of neural networks as optimizable black-box functions. We start by asking the question: *What "physical relations" can we model beyond next-state prediction?*

Fig. 11 compares three classical approaches to modeling physical systems without machine learning:

- Case (I): Global analytical solution. For simple systems with regular structures, one can derive a closed-form solution directly.

- Case (II): PDE + numerical integration. In more complex settings where no closed-form solution exists, the standard practice is to formulate the system's dynamics as a PDE and solve it step-by-step over time via numerical methods. This local integration approach underlies most physics-constrained neural network designs that incorporate the PDE operators into the network.

- Case (III): Direct global relation. In some complex systems, states that are temporally far apart can be related directly using global conservation laws (*e.g.*, energy conservation). This is akin to high-school physics problems: one can compute an object's velocity at a certain position from initial conditions alone, without solving for a full trajectory. While this is less general than PDE-based approaches, it suggests a promising avenue: leveraging global physical principles within a black-box neural network could extend this technique to more complex, real-world dynamical systems beyond simple textbook problems.

Prior works such as HNN Greydanus et al. (2019) draw inspiration from the intuition of Case (III) to leverage global conservation laws, but are still realized in the form of local, step-by-step constraints between adjacent time steps, as in Case (II). To move a step forward, we aim to design network architectures that enforce global conservation principles in a truly global manner, allowing black-box neural networks to capture complex, long-range, and indirect physical relationships. This approach seeks to fully exploit the expressive power of neural networks while remaining faithful to fundamental physical laws.

# B  Discrete Left Hamiltonian $H^-$

The discrete right Hamiltonian $H^-$ gives the equation of motion in the form

$$q_t = -\nabla_p H^-(q_{t+1}, p_t), \quad p_{t+1} = -\nabla_q H^-(q_{t+1}, p_t). \tag{9}$$

It can be a first-order approximation of the continuous Hamiltonian $\mathcal{H}$ by

$$q_t = q_{t+1} - \Delta t \nabla_p \mathcal{H}(q_t, p_{t+1}), \quad p_{t+1} = p_t - \Delta t \nabla_q \mathcal{H}(q_t, p_{t+1}). \tag{10}$$

When the states are extended to blocks, the block-wise discrete left Hamiltonian is defined as

$$Q_t^{t+b} = -\nabla_P H^-(Q_{t+s}^{t+s+b}, P_t^{t+b}), \quad P_{t+s}^{t+s+b} = -\nabla_Q H^-(Q_{t+s}^{t+s+b}, P_t^{t+b}). \tag{11}$$

Fig. 12 below illustrates the relation between discrete left and right Hamiltonians in both classical forms and our block-wise extensions. Both the left and right Hamiltonians take each other's outputs as inputs.

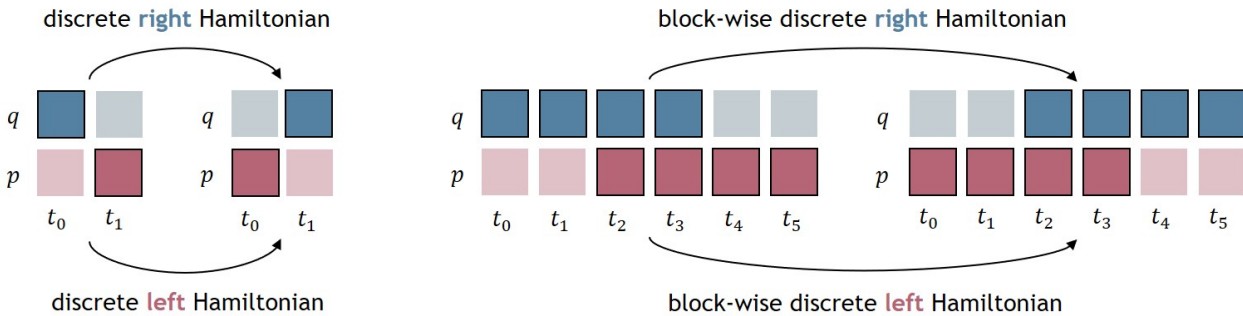

Figure 12: **Discrete left and right Hamiltonian blocks.** They describe the same formulation of a physical system with different discretizations (that both preserve the system's symplectic structures). They take each other's outputs as inputs and vice versa.

## C  Physical Interpretations for DHN

In this section, we discuss whether extending the discrete Hamiltonian to block sizes and strides greater than 1 still allows for explicit physical interpretations. Specifically, we address the following two questions:

(i) *What is the conserved quantity with the block-wise Hamiltonian?* For a discrete Hamiltonian block of size $b$, the conserved quantity is the sum of the total energy of $b$ independent states. More specifically, the states within a discrete Hamiltonian block can be interpreted as those of identical physical systems, each starting at a different time. Fig. 13 provides an illustration of this concept.

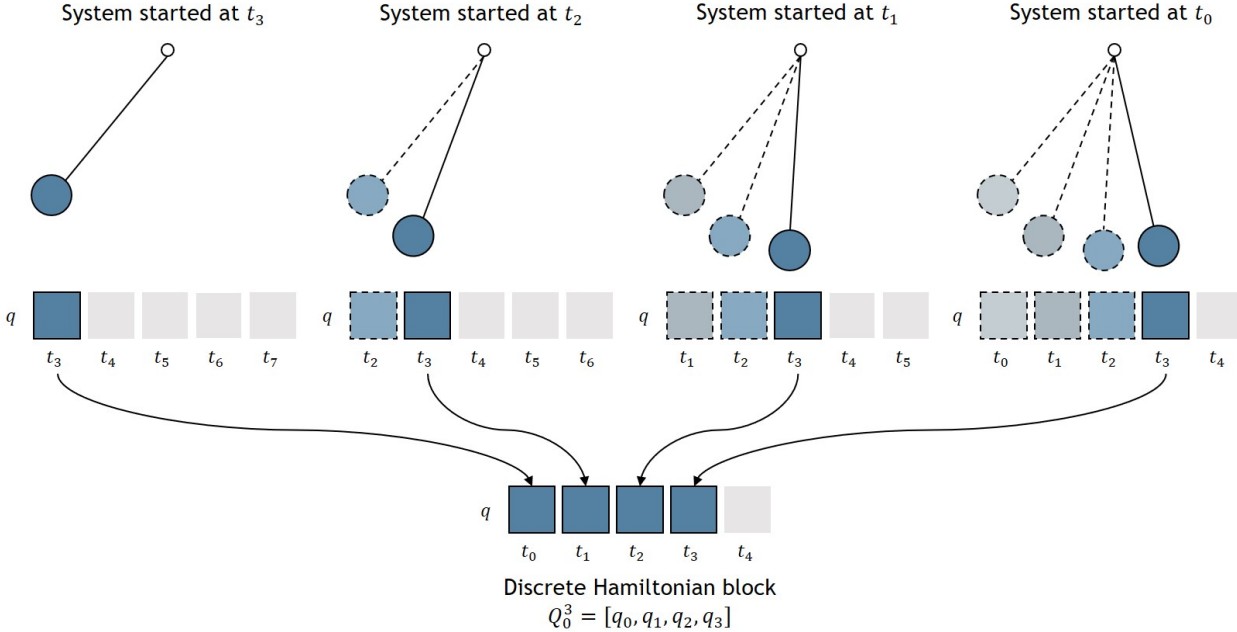

Figure 13: **Physical interpretations of block-wise discrete Hamiltonian.** The figure shows an example of block size $b = 4$. The states within a discrete Hamiltonian block can be interpreted as those of identical physical systems, each starting at a different time step.

Consider the case where the block size is $b = 4$. Suppose we have four identical physical systems, each initialized at different times: $t_0, t_1, t_2, t_3$. By time $t_3$, these systems will have evolved for 0, 1, 2, and 3 time steps, respectively. If we take their states at $t_3$ and stack them together, we obtain a state block that effectively represents four consecutive states spanning four time steps within a single system. Importantly, the four states at $t_3$ remain independent, as the four duplicated systems do not interact with one another. Thus, the conserved quantity in this framework is the total energy summed across all these identical, non-interacting systems. Fig. 14 provides a detailed illustration of this.

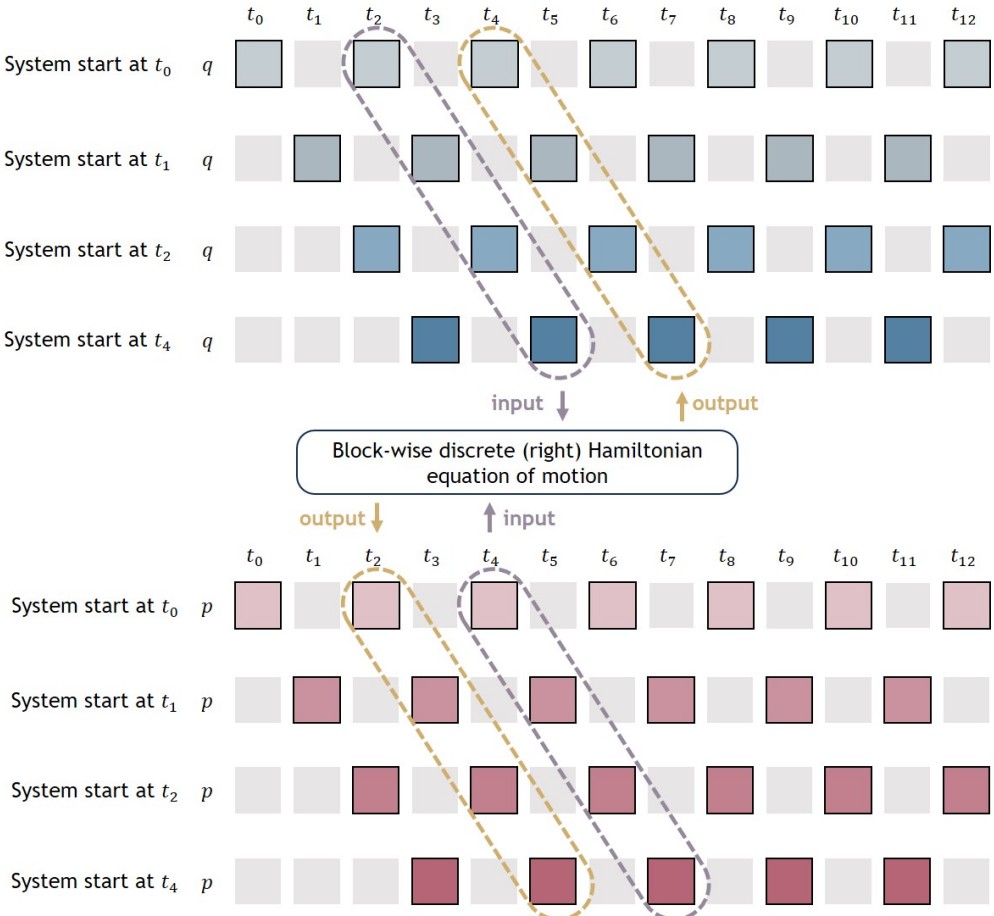

Figure 14: **Detailed explanations of the block-wise discrete Hamiltonian.** The figure shows an example of block size $b = 4$ and stride $s = 2$.

(ii) *What are the relaxations compared to the classical discrete Hamiltonian?* When extending the classical discrete Hamiltonian to a block-wise formulation, certain physical constraints are relaxed. The two main relaxations are as follows:

First, instead of conserving the energy of each individual state, the block-wise Hamiltonian conserves the total energy summed over $b$ states. This allows for different energy distributions across the $b$ states, making the constraint weaker than enforcing per-state energy conservation.

Second, as discussed in Sec. 4.2, when the stride $s$ is smaller than the block size $b$, there is an overlap of $b - s$ between network inputs and outputs. In theory, exact energy conservation (in the generalized form) requires that the overlapping states remain identical. However, in practice, this self-consistency loss is rarely minimized to exactly zero. The extent to which it is minimized depends on factors such as network expressivity, architecture, and hyperparameters $b$ and $s$, which in turn affect how well energy conservation is maintained.

Despite these relaxations, the model still enforces a form of physical consistency across the trajectory. Rather than strictly conserving per-state energy, it shifts toward preserving higher-level conserved quantities. This relaxation also opens the door to developing more abstract notions of physical consistency on latent embeddings instead of the raw observed states.

## C.1 Formal Derivations of the Conserved Quantity

Here we provide formal derivations of the (generalized) energy conservation in our block-wise Hamiltonian. We show that after extending the discrete Hamiltonian into our block-wise discrete Hamiltonian with block size $b$ and stride $s$, it still has a conserved quantity that relates to the system energy.

The discrete Hamiltonian can be derived in two ways: on the one hand, it can be directly derived from discrete systems, and the energy conservation (time invariance) is expressed as its symplectic structure (intuitively, it means that across a long time, the energy fluctuate around a constant value, instead of staying constant precisely); on the other hand, it can be derived as the first-order approximation of the continuous Hamiltonian, and the energy conservation also refers to approximately conserving the energy of continuous system. Here, for our block-wise Hamiltonian, we analyze its conserved quantity with the second interpretation, because the symplectic structure is highly relevant to numerical integrations for state updates from $t$ to $t+1$, while in our case with stride $s > 1$ and different masking patterns, our state update is much more flexible.

**First-order approximation of the continuous Hamiltonian**   Given time steps $t = 0, \cdots, T$, suppose we have a physical trajectory with states $(q_0, p_0), (q_1, p_1), \cdots, (q_T, p_T)$. The total action with the discrete right Hamiltonian is

$$S^+ = \sum_{t=0}^{T-1} (p_{t+1} \cdot q_{t+1} - H^+(q_t, p_{t+1})). \tag{12}$$

On the other hand, the action of the continuous Hamiltonian is

$$\mathcal{S} = \int_0^1 (p(\tau) \cdot q(\tau) - \mathcal{H}(q(\tau), p(\tau))) \mathrm{d}\tau. \tag{13}$$

Here we assume that the discrete time steps are uniformly sampled between $[0, 1]$ with interval $\Delta t = 1/T$. To derive numerical integrators using discrete Hamiltonians, it is important to recognize that the discrete action sum must approximate the continuous action Lall & West (2006); Marsden & West (2001); Hairer et al. (2006), that is, $S^+ \approx \mathcal{S}$. This means that the right Hamiltonian should satisfy

$$H^+(q_t, p_{t+1}) \approx p_{t+1} \cdot q_{t+1} - \int_{t\Delta t}^{(t+1)\Delta t} (p(\tau) \cdot \dot{q}(\tau) - \mathcal{H}(q(\tau), p(\tau))) \, \mathrm{d}\tau. \tag{14}$$

With first-order approximation, we have

$$
\begin{aligned}
&H^+(q_t, p_{t+1}) \\
&= p_{t+1} \cdot (q_t + \Delta t \nabla_p \mathcal{H}(q_t, p_{t+1})) - \Delta t(p_{t+1} \nabla_p \mathcal{H}(q_t, p_{t+1}) - \mathcal{H}(q_t, p_{t+1}))) + \mathcal{O}(\Delta t^2) \\
&= p_{t+1} \cdot q_t + \Delta t \mathcal{H}(q_t, p_{t+1}) + \mathcal{O}(\Delta t^2).
\end{aligned} \tag{15}
$$

With time invariance, the continuous Hamiltonian $\mathcal{H}$ conserves the total energy of the system, and the discrete right Hamiltonian $H^+$ is an approximation of that.

**Block-wise approximation**   Consider a discrete right Hamiltonian block of block size $b$ and stride $s$, with inputs are $Q_t^{t+b} = [q_t, \cdots, q_{t+b}]$ and $P_{t+s}^{t+s+b} = [p_{t+s}, \cdots, p_{t+s+b}]$ and satisfying the equation of motion as in Eq. 5 for any $t$. The total action approximation generalizes to

$$H^+(Q_t^{t+b}, P_{t+s}^{t+s+b}) \approx \sum_{k=t}^{t+b} \left( p_{k+s} \cdot q_{k+s} - \int_{k\Delta t}^{(k+s)\Delta t} (p(\tau) \cdot \dot{q}(\tau) - \mathcal{H}(q(\tau), p(\tau))) \, \mathrm{d}\tau \right). \tag{16}$$

And similar to Eq. 15, we have

$$H^+(Q_t^{t+b}, P_{t+s}^{t+s+b}) = \sum_{k=t}^{t+b} (p_{k+s} \cdot q_k + s\Delta t \mathcal{H}(q_k, p_{k+s})) + \mathcal{O}(\Delta t^2). \tag{17}$$

Omitting the higher-order terms $\mathcal{O}(\Delta t^2)$, this can be viewed as a summation of $b$ identical systems starting at time steps $t = 0, \cdots, b$, and their continuous Hamiltonians are discretized with time intervals $s\Delta t$. In this analogy, the total conserved quantity is an approximation of

$$\sum_{k=0}^{b} \mathcal{H}(q(\tau + k\Delta t), p(\tau + k\Delta t)). \tag{18}$$

## D Denoising Inference

As mentioned in Sec. 3.3, at training time, we apply noises with randomly sampled scales to different unknown states. However, at inference time, we progressively denoise the unknown states with a sequence of decreasing noise scales that are synchronized on all unknown states. Fig. 15 illustrates the iterative denoising process at inference time with a pair of DHN blocks $H^+$ and $H^-$.

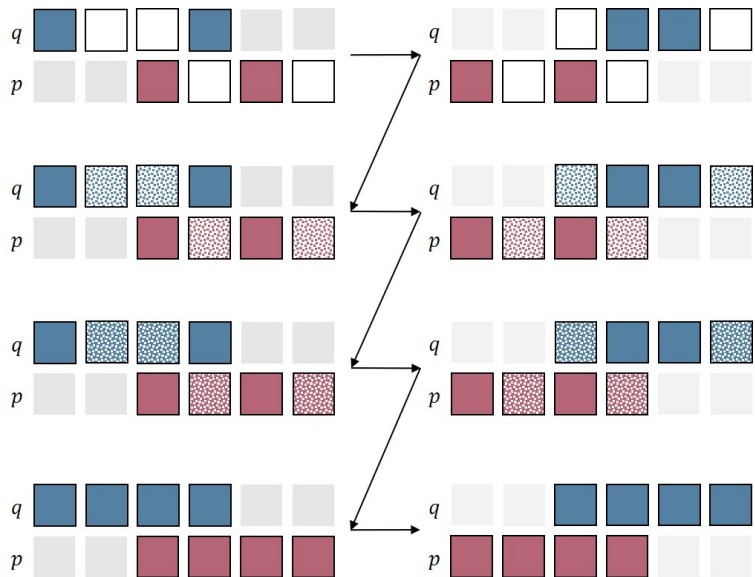

Figure 15: **Iterative denoising at inference time.** A pair of DHN blocks, $H^+, H^-$, with block size $b$ and stride $s$, are jointly applied to a stack of $b + s$ states to denoise the unknown blocks. Progressive denoising is conducted by iteratively applying block-wise $H^+$ and $H^-$ and gradually decreasing the noise scales.

Consider a block of states $(Q_t^{t+s+b}, P_t^{t+s+b})$ with some unknown slots indicated by binary masks $(M_t^{t+s+b}, N_t^{t+s+b})$. We iterative apply $H^+(Q_t^{t+b}, P_{t+s}^{t+s+b})$ and $H^-(Q_{t+s}^{t+s+b}, P_t^{t+b})$ to infer the unknown states conditioned on the known states. We sample the initial state predictions from a Gaussian distribution and then apply iterative denoising with decreasing noise levels $1 = \alpha_N > \alpha_{N-1} > \cdots > \alpha_1 > \alpha_0 = 0$, as in Algorithm 1. This is similar to the diffusion models Ho et al. (2020); Song et al. (2020).

## E Network Architecture Details

As mentioned in Sec. 3.4, we employ a decoder-only transformer to learn the block-wise Hamiltonian. Its detailed architecture is shown in Fig. 16. The network is very lightweight, with only two transformer blocks, plus input embeddings and the (linear) output projection. The two transformer blocks are identical, with input/output feature dimension 128, FFN hidden feature dimension 120×4, and 4 heads in their multi-head attentions.

---

**Algorithm 1:** Iterative denoising with the Hamiltonian blocks

---

**Data:** Known states $(Q_{\text{known}})_t^{t+s+b}, (P_{\text{known}})_t^{t+s+b}$, binary state mask $(M_t^{t+s+b}, N_t^{t+s+b})$.
**Result:** The complete block $Q_t^{t+s+b}, P_t^{t+s+b}$ with the masked states predicted.
Sample initial state predictions $Q_t^{t+s+b}, P_t^{t+s+b} \sim \mathcal{N}(0, \mathbf{I})$.
**for** $n = N - 1, \cdots, 0$ **do**

Sample noise $\mathcal{E}, \mathcal{E}' \sim \mathcal{N}(0, \mathbf{I})$, and apply $H^+$ denoising:

$$(Q_{\text{pred}})_{t+s}^{t+s+b} \leftarrow (1 - \alpha_n)\nabla_P H^+(Q_t^{t+b}, P_{t+s}^{t+s+b}) + \alpha_n \mathcal{E},$$
$$(P_{\text{pred}})_t^{t+b} \leftarrow (1 - \alpha_n)\nabla_Q H^+(Q_t^{t+b}, P_{t+s}^{t+s+b}) + \alpha_n \mathcal{E}'.$$

Combine the predicted states with the known states:

$$Q_{t+s}^{t+s+b} \leftarrow (1 - M_{t+s}^{t+s+b})(Q_{\text{pred}})_{t+s}^{t+s+b} + M_{t+s}^{t+s+b}(Q_{\text{known}})_{t+s}^{t+s+b},$$
$$P_t^{t+b} \leftarrow (1 - M_t^{t+b})(P_{\text{pred}})_t^{t+b} + M_t^{t+b}(P_{\text{known}})_t^{t+b}.$$

Sample noise $\mathcal{E}, \mathcal{E}' \sim \mathcal{N}(0, \mathbf{I})$, and apply $H^-$ denoising:

$$(Q_{\text{pred}})_t^{t+b} \leftarrow -(1 - \alpha_n)\nabla_P H^-(Q_{t+s}^{t+s+b}, P_t^{t+b}) + \alpha_n \mathcal{E},$$
$$(P_{\text{pred}})_{t+s}^{t+s+b} \leftarrow -(1 - \alpha_n)\nabla_Q H^-(Q_{t+s}^{t+s+b}, P_t^{t+b}) + \alpha_n \mathcal{E}'.$$

Combine the predicted states with the known states:

$$Q_t^{t+b} \leftarrow (1 - M_t^{t+b})(Q_{\text{pred}})_t^{t+b} + M_t^{t+b}(Q_{\text{known}})_t^{t+b},$$
$$P_{t+s}^{t+s+b} \leftarrow (1 - M_{t+s}^{t+s+b})(P_{\text{pred}})_{t+s}^{t+s+b} + M_{t+s}^{t+s+b}(P_{\text{known}})_{t+s}^{t+s+b}.$$

**end**

---

## F  Experimental Setups

### F.1  Physical Systems and Parameter Randomization

Here we elaborate on the details of the two settings we experiment with: the *single pendulum* and the *double pendulum*, as illustrated in Fig. 17. In both settings, we first define the generalized coordinate $q$ and the system's Lagrangian $\mathcal{L}(q, \dot{q})$. The generalized momenta is then defined by $p = \nabla_{\dot{q}} \mathcal{L}$. We set the gravitational acceleration $g = 0.981$.

**Single pendulum** In this system, the varied parameter is the string length $l$, randomly sampled between $[0.5, 1.0]$ for each trajectory. The mass of the ball is set to be $m = 1$. The generalized coordinate is defined as $q = \theta$, with initial value $\theta = \pi/2$ for all trajectories. The Lagrangian of the system is

$$\mathcal{L} = \frac{1}{2}ml^2\dot{q}^2 - mgl(1 - \cos q). \tag{19}$$

Here $(q, p)$ are the standard angular position and angular momentum in spherical coordinates.

**Double pendulum** In this system, the varied parameter is the string length $l_2$, randomly sampled between $[0.5, 1.5]$ for each trajectory. The remaining fixed parameters are $l_1 = 1, m_1 = m_2 = 1$. The generalized coordinate is defined as $q = (\theta_1, \theta_2)$, with initial values $\theta_1 = \theta_2 = \pi/2$ for all trajectories. The Lagrangian of the system is

$$\mathcal{L} = \frac{1}{2}(m_1 + m_2)l_1^2\dot{\theta}_1^2 + \frac{1}{2}m_2 l_2^2\dot{\theta}_2^2 + m_2 l_1 l_2 \dot{\theta}_1\dot{\theta}_2 \cos(\theta_1 - \theta_2) + (m_1 + m_2)gl_1\cos\theta_1 + m_2 gl_2\cos\theta_2. \tag{20}$$

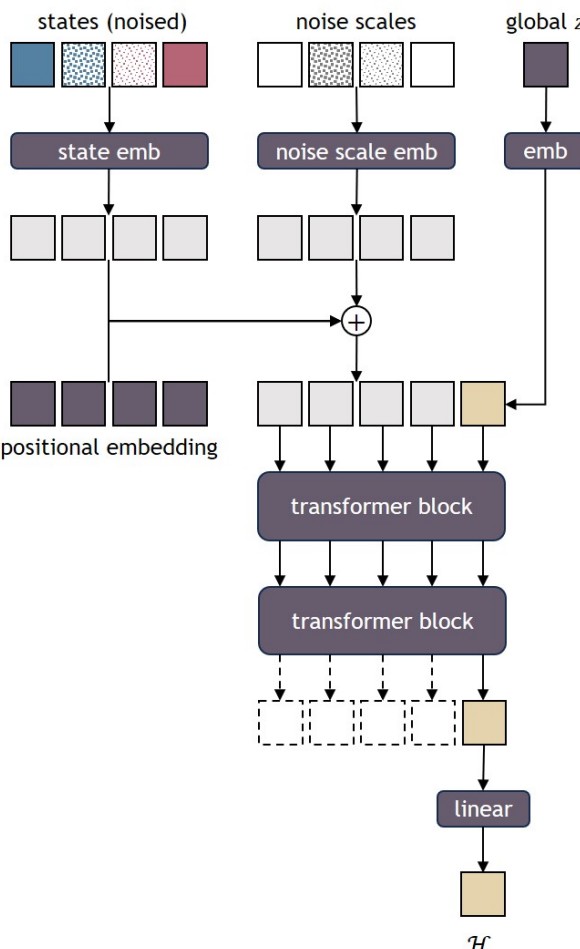

Figure 16: **Detailed architecture of the decoder-only transformer.** Learnable modules or variables are shown in dark purple. The final output of the network is a scalar Hamiltonian value, decoded from the position of the global latent code in the transformer (shown in yellow).

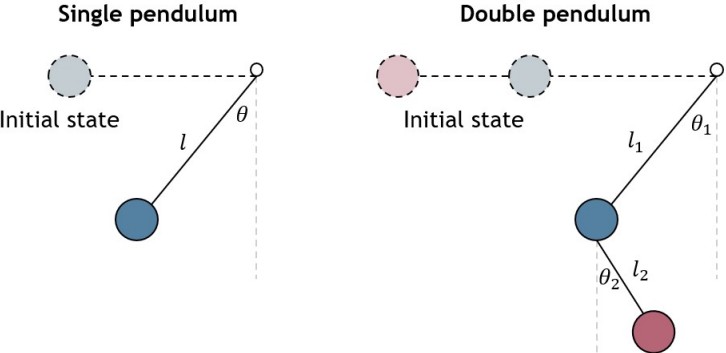

Figure 17: **Physical systems for the experiments.** Circles with dotted lines and swallower colors are the initial states, which are identical to all training and testing trajectories. Circles with solid lines and darker colors illustrate the intermediate states along the simulated trajectory.

### F.2 Network Training and Computational Resources

At the training stage, we optimize for both the network weights and the latent codes. We use an Adam optimizer with a learning rate $10^{-4}$ and weight decay $10^{-4}$ and train for 200 epochs with a batch size of 32.

At the test-time optimization stage, we freeze the network weights and only optimize for the latent codes. e use an Adam optimizer with a learning rate $10^{-2}$ and weight decay $10^{-4}$. As all latent codes are optimized independently, we optimize for 1000 epochs with a batch size of 100.

All networks are trained on a single Titan RTX 24GB GPU.

