# OpenReview forum: "Denoising Hamiltonian Network for Physical Reasoning"
_TMLR — Accepted by TMLR_

### Review · Reviewer_3Acm · 2025-10-22

**Summary Of Contributions:**

This paper explores what physical attributes a neural network should possess when modeling dynamical systems. While partial differential equations describe systems through strictly local temporal relations, neural networks need not be bound by this locality. The Denoising Hamiltonian Network extends traditional Hamiltonian neural operators to capture non-local, block-wise temporal interactions while maintaining core physical constraints such as energy conservation. By introducing learnable block sizes, variable strides, and a denoising objective, Denoising Hamiltonian Network generalizes physical reasoning beyond forward simulation, enabling multi-scale temporal modeling and robustness against numerical integration errors. This framework bridges the gap between rigid physical formalism and the flexibility of deep networks, offering a new paradigm for physics-based learning architectures.

**Audience:**

Yes

**Audience Explanation:**

Yes, the findings of this paper would likely interest a portion of TMLR’s audience, particularly those working on temporal modeling, scientific machine learning, and sequence prediction. The model in this paper introduces a block-wise operator that generalizes beyond traditional one-step (one-tick) prediction, enabling reasoning over multi-tick temporal blocks. This represents a conceptual shift similar to what is now emerging in broader machine learning fields — moving from local, incremental updates toward non-local, span-level modeling that captures higher-order dependencies.

**Claims And Evidence:**

Yes

**Claims Explanation:**

The claims made in the submission are partially supported by accurate and convincing evidence. The paper provides solid experimental results on representative tasks such as forward trajectory prediction and super-resolution interpolation, demonstrating that the proposed Denoising Hamiltonian Network (DHN) can capture non-local temporal dynamics and outperform baselines in long-term stability and reconstruction accuracy. The methodology is clearly presented, and the empirical trends align with the theoretical motivation for block-wise and denoising-based physical reasoning.

**Requested Changes:**

1. Include missing examples: The paper mentions parameter estimation and trajectory inpainting as representative reasoning tasks but does not provide corresponding experimental demonstrations. Including at least one clear example—such as parameter estimation on a pendulum or spring system—would make the claims of general applicability more convincing and illustrate DHN’s versatility beyond forward prediction and interpolation.
2. Improve figure clarity and discuss figure implications: Figures 6 and 7, which are central to illustrating DHN’s comparative performance and stability, are difficult to interpret due to small fonts and overly similar color choices. The visual distinctions between models and baselines should be enhanced, and key trends or takeaways should be clearly annotated within the figures. A more legible design would help readers better appreciate DHN’s improvements in long-term trajectory reconstruction and error reduction

---

> ### Author Response · Authors · 2025-11-17
>
> We thank the reviewer for the feedback and constructive suggestions.
>
> **Missing examples: parameter estimation experiments.**
>
> Our paper already includes an example of parameter estimation in Section 4.2, where we use self-supervised representation learning with linear probing to predict the string length in the double pendulum. We chose this representation-learning setup (rather than direct end-to-end supervised parameter estimation) for two reasons: (i) random-masking–based representation learning has shown remarkable success in language and vision (e.g., BERT [1] and MAE [2]), and we were interested in exploring whether similar mechanisms can learn latent physical properties; and (ii) physical representation learning remains relatively underexplored compared to both representation learning for semantic or visual attributes and fully supervised physical parameter estimation, and thus we hope this experiment helps draw more attention to the problem of learning physical representations.
>
> [1] Devlin, J., Chang, M.W., Lee, K. and Toutanova, K., 2019, June. Bert: Pre-training of deep bidirectional transformers for language understanding. In Proceedings of the 2019 conference of the North American chapter of the association for computational linguistics: human language technologies, volume 1 (long and short papers) (pp. 4171-4186).
>
> [2] He, K., Chen, X., Xie, S., Li, Y., Dollár, P. and Girshick, R., 2022. Masked autoencoders are scalable vision learners. In Proceedings of the IEEE/CVF conference on computer vision and pattern recognition (pp. 16000-16009).
>
> **Improve figure clarity.**
>
> Thank you for the suggestion. We will improve Figures 6 and 7 by increasing font sizes, adjusting colors for clearer distinction, and adding annotations to better highlight the key trends and the differences between DHN and the baselines.

---

### Review · Reviewer_SNKY · 2025-11-03

**Summary Of Contributions:**

This work proposes Denoising Hamiltonian Networks (DHNs) that incorporate a denoising loss in addition to regular Hamiltonian loss over block data. The benefits include better generalization over multiple trajectories in a given system and improved long-range modeling.

**Audience:**

Yes

**Audience Explanation:**

I believe this work will interest the ML research community that focuses on physical reasoning using machine learning.

**Broader Impact Concerns:**

No concerns about the ethical implications.

**Claims And Evidence:**

Yes

**Claims Explanation:**

From my understanding, the claim is that a Hamiltonian neural network (HNN) with denoising can improve generalization and long-range modeling of physical systems. These metrics are improved by the proposed denoising Hamiltonian neural network (DHN) in the experimental results. However, the experiments do not clearly show that the improvements actually come from the architectural/training choices, and not the nature of the data used. Therefore, I feel that, currently, the claims are partially supported.

**Requested Changes:**

The following are a mix of requested changes and questions about the work. The important ones have a hash symbol (#) next to their numbering. My major concerns are with writing -- specifically, in comparing the contributions of this work against recently proposed works and clearly explaining the experimental design choices.

**# C1. Related works**: The absence of recent related works in this submission makes it hard to gauge the contribution of the paper. Only three references from 2021-present are included in this work, and none of them are related to the core topic of this work. Although the submission is evaluated on the basis of claim-and-evidence, it is difficult to know how original the primary claim in this paper is without a sufficient literature review. The lack of recent related works particularly impacts the introduction section. The introduction section currently includes no references, and the second paragraph that discusses the limitations of current methods is on weak grounds with references. If the limitations listed in the second paragraph cannot be backed with references, then the existence of these limitations must be proven in the work.

**# C2. Role of denoising**: Why is denoising required? Sec. 3.3 (first paragraph) says "we want the Hamiltonian blocks to...also learn the state optimization per time step for inference." I didn't understand why the Hamiltonian blocks must learn state optimization for inference and how denoising helps to achieve that goal. Is there an ablation on what happens without $\mathcal{L}_{\text{denoise}}$?

**# C3. Noisy samples for HNN**: This question is related to C2. Were noisy samples used in training HNNs (baselines)? I know HNNs do not have a denoising model, but I was wondering if noisy training data implicitly improves generalization. If it doesn't improve the performance, then this experiment will also strengthen the cause for $\mathcal{L}_{\text{denoise}}$.

**# C4. Role of overlapping regions**: The baselines are trained on "pairs of adjacent time steps," while DHNs are trained on overlapping blocks. How much difference will this make to the performance? Please correct me if it's not possible to compare these two settings.

**C5. Parameter estimation experiments**: There are neural network-based methods for parameter estimation in PDEs from a few measurement samples [R6, R7]. Are these baselines valid for the experiments in Sec. 4.2?

**C6. Why is $z$ included?** Why is the global latent code $z$ included in the architecture? Can the authors clarify how it "serves as a query token for outputting the Hamiltonian value?"

**C7. Super-resolution experiment baseline**: How is the CNN used for super-resolution? And why is a CNN without a physical prior used for trajectory interpolation, instead of cubic Hermite splines or physics-informed learned models?

**C8. Usage of the term "neural operator"**: In the second paragraph of the introduction, it is written "we aim to design more general neural operators that both follow physical constraints, and unleash the expressivity of neural networks..." It is not clear how "neural operator" is different from "neural networks" in the submitted work. The term "neural operator" is generally used in contemporary works [R1-4] with its meaning introduced by [R5].

**References**:

[R1] Ziming Liu, Yixuan Wang, Sachin Vaidya, Fabian Ruehle, James Halverson, Marin Soljačić, Thomas Y. Hou, Max Tegmark, "KAN: Kolmogorov-Arnold Networks", ICLR 2025.

[R2] Boris Bonev, Thorsten Kurth, Christian Hundt, Jaideep Pathak, Maximilian Baust, Karthik Kashinath, Anima Anandkumar, "Spherical Fourier Neural Operators: Learning Stable Dynamics on the Sphere", ICML 2023.

[R3] Makoto Takamoto, Timothy Praditia, Raphael Leiteritz, Dan MacKinlay, Francesco Alesiani, Dirk Pflüger, Mathias Niepert, "PDEBENCH: An Extensive Benchmark for Scientific Machine Learning", NeurIPS (D&B) 2022.

[R4] Steven L. Brunton, J. Nathan Kutz, "Promising directions of machine learning for partial differential equations", Nature Computational Science, 2024.

[R5] Nikola Kovachki, Zongyi Li, Burigede Liu, Kamyar Azizzadenesheli, Kaushik Bhattacharya, Andrew Stuart, Anima Anandkumar, "Neural Operator: Learning Maps Between Function Spaces", JMLR 2023.

[R6] Masanobu Horie, Naoto Mitsume, "Physics-Embedded Neural Networks: Graph Neural PDE Solvers with Mixed Boundary Conditions", NeurIPS 2022.

[R7] Xuyang Li, Hamed Bolandi, Talal Salem, Nizar Lajnef, Vishnu Naresh Boddeti, "NeuralSI: Structural Parameter Identification in Nonlinear Dynamical Systems", ECCV 2022 Workshop on Computer Vision for Civil and Infrastructure Engineering.

---

> ### Author Response · Authors · 2025-11-17
>
> We would like to begin by sharing more of the motivation behind our work, as this may help clarify several points where our perspective differs from the traditional scientific/physical learning viewpoint. Broadly speaking, current approaches tend to fall into two extremes: (i) **strict physics-driven operators** that enforce physical structure by construction (e.g., equivariant architectures, Hamiltonian networks), which provide strong guarantees but often have limited scalability; and (ii) **pure black-box neural networks**, which are highly expressive and scalable but typically capture semantic patterns rather than physical structure. Our aim is to explore the underdeveloped space in between these extremes: designing neural operators that incorporate physical constraints while maintaining the flexibility and expressivity of modern deep networks.
>
> At the same time, our methodological perspective is influenced strongly by computer vision and representation learning, where the emphasis is typically on *semantics*, *generalization*, and discovering latent structure across diverse data. This contrasts with much of the PDE-oriented physical learning literature, where the goal is often to recover a single governing equation with high fidelity (overfitting to one system). In this work, we intentionally study a different class of problems that we believe are underexplored in physical learning: extrapolating to unseen system parameters, learning representations from *collections* of trajectories rather than solving a single equation, and developing latent codes that capture higher-level system structure. To tackle these challenges, we adopt a “non-typical” physical-learning framework that draws inspiration from computer vision/graphics. Our intention is not to replace existing PDE-based approaches, but to broaden the range of questions and methodologies considered in physical learning by bringing in complementary ideas from other areas.
>
> We kindly suggest that the reviewer consider our method and the responses below through this broader interdisciplinary perspective. More concrete answers to questions C1–C8 are provided next

---

> ### Author Response · Authors · 2025-11-17
> **C1. Related works**
>
> **C1. Related works.**
>
> The term “physics-informed neural network (PINN)” is now very broad, covering many types of works on diverse problems as the community grows. Our work focuses on a particular aspect of physical learning: balancing *generalization* across systems with *physical guarantees*. While the PINN literature is extensive, much of it focuses on fitting a single system or solving a specific PDE, with limited attention to generalization across a collection of related systems. For example, [R2] trains a network to solve one PDE on a sphere, which we consider “single-system fitting.” In contrast, our work asks: if we train on 1,000 different systems of similar form, can we predict solutions or parameters for an *unseen* system?
>
> From a methodological perspective, works on equivariant or guaranteed networks are more relevant, as they also explore which constraints can be built into network architectures and which can be relaxed. Most of these works are not specifically designed for physics applications, which is why they were not discussed in the original paper. Nevertheless, we recognize that these connections help contextualize our contributions and will include them in the revision.
>
> While the reviewer notes the limited number of recent references, our focus was on works most relevant to the problem studied rather than strictly on recent publications. To clarify and strengthen the paper, we will add discussions of these related works and explicitly cite recent papers on both physics-informed learning and equivariant networks. We hope this contextualization makes our contributions clearer and highlights the novelty of addressing generalization in a physically constrained framework.

---

> ### Author Response · Authors · 2025-11-17
> **C2. Role of denoising**
>
> **C2. Role of denoising.**
>
> The denoising process in our framework is designed to construct an *iterative inference strategy* within the network. An ablation is not possible in this case, as the network cannot function without it: all the states we care about (the $q$'s and $p$’s, both known and unknown) are **inputs**, not outputs of the network. At inference time, the unknown states are initialized randomly and iteratively updated. This strategy is common in deep learning frameworks that rely on iterative inference, such as diffusion models [1, 2], auto-decoding methods like DeepSDF [3], test-time training (TTT) [4], etc.
>
> The motivation for introducing denoising is to enable the network to perform iterative refinement, similar to optimization in variational integration. Variational integration can achieve higher precision than standard forward methods, but it requires Hamiltonian evaluations not only along the trajectory but also at points off the trajectory (“free space”) to gradually optimize unknown states towards the true trajectory. In practice, deep learning models only have access to the observed trajectories as training data; acquiring full information in free space would greatly increase data requirements.
>
> To address this, we draw inspiration from diffusion models: by introducing handcrafted noise into the network, we effectively “simulate” optimization trajectories during training. This design embeds the iterative inference process directly into the network, allowing it to refine state estimates without requiring additional true data. For more details on the role of this noise, please see our response to C3.
>
> [1] Ho, J., Jain, A. and Abbeel, P., 2020. Denoising diffusion probabilistic models. Advances in neural information processing systems, 33, pp.6840-6851.
>
> [2] Song, Y., Sohl-Dickstein, J., Kingma, D.P., Kumar, A., Ermon, S. and Poole, B., 2021. Score-based generative modeling through stochastic differential equations.  International Conference on Learning Representations (ICLR).
>
> [3] Park, J.J., Florence, P., Straub, J., Newcombe, R. and Lovegrove, S., 2019. Deepsdf: Learning continuous signed distance functions for shape representation. In Proceedings of the IEEE/CVF conference on computer vision and pattern recognition (pp. 165-174).
>
> [4] Sun, Y., Wang, X., Liu, Z., Miller, J., Efros, A.A. and Hardt, M., 2019. Test-time training for out-of-distribution generalization.

---

> ### Author Response · Authors · 2025-11-17
> **C3. Noisy samples**
>
> **C3. Noisy samples.**
>
> The “noise” in our denoising Hamiltonian framework is not related to adding noise to the training or test data. Instead, it is analogous to the noise in denoising diffusion models: it enables an iterative process that progressively refines the predictions. This is an architectural design choice built into the learning framework, rather than a mechanism for improving robustness or generalization to noisy observations.
>
> To draw the analogy, diffusion models are typically trained on clean images with no added noise; the noise in the model is handcrafted and independent of the data, serving only to define a progressive inference process. To note, in this analogous task of image generation, the same task can be solved without any handcrafted noise (for example, using an autoregressive model), demonstrating that the noise is not intrinsic to the problem itself.
> Thus, the noise in our method should not be interpreted as a data augmentation or generalization technique – it is a part of the iterative refinement process in the architecture.
>
> [1] Ho, J., Jain, A. and Abbeel, P., 2020. Denoising diffusion probabilistic models. Advances in neural information processing systems, 33, pp.6840-6851.
>
> [2] Song, Y., Sohl-Dickstein, J., Kingma, D.P., Kumar, A., Ermon, S. and Poole, B., 2021. Score-based generative modeling through stochastic differential equations.  International Conference on Learning Representations (ICLR).

---

> ### Author Response · Authors · 2025-11-17
> **C4. Role of overlapping regions**
>
> **C4. Role of overlapping regions.**
>
> The overlapping region is essential for our method, especially for forward simulation under the masking pattern used for unknown future states. Without overlap, a block would take the form $H(q_1, \cdots, q_b, p_{b+1}, \cdots, p_{2b})$. And masking out the second half would give the network an input of $H(q_1, \cdots, q_b, \text{noise}, \cdots, \text{noise})$.
> In this setting, the denoising process would need to infer all momenta $p_1, \cdots, p_b$ from the positions $q_1, \cdots, q_b$. This is fundamentally ill-posed: for a given set of positions, there are infinitely many valid momenta depending on the system’s initial conditions. By ensuring that at least one $(q_t, p_t)$ pair is observed within each block, the overlapping region removes this ambiguity and makes the inference tractable.
> HNNs do not encounter this issue because they never need to iteratively infer **input states** -- their inputs are always fully specified within the framework of a forward integrator (e.g., forward Euler).
>
> Conceptually, the overlapping region plays a role analogous to the **stride** in ConvNets: changing the stride affects how much contextual continuity is preserved between consecutive blocks. It also connects directly to the relaxation of conservation laws, as discussed in **Appendix C**, where energy conservation in the generalized Hamiltonian sense relies on maintaining identical inputs/outputs on the overlapping region.
>
> Empirically, the influence of the overlap is quantified in **Figure 8 right**. Varying block sizes and strides changes the amount of overlap (i.e.,$b-s$). Larger overlaps strengthen self-coherence but reduce the emphasis on learning relations across more distant states, while smaller overlaps encourage longer-range inference but weaken self-consistency and stability. The extremes are illustrative:
> - Zero stride (full overlap) collapses the task into enforcing identical inputs and outputs.
> - Zero overlap (as in HNN) removes the structural coupling that supports stable block-wise inference.
>
> The results indicate that moderate overlaps (approximately $s \approx b/2$) offer the best balance.
>
> Because the overlapping region is functionally necessary for resolving momenta ambiguity and structurally important for stabilizing block-wise Hamiltonian inference, it is not directly meaningful to compare a “no-overlap” version of DHN to baselines that do not rely on iterative denoising. This difference is inherent to the architectural design rather than a hyperparameter choice.

---

> ### Author Response · Authors · 2025-11-17
> **C5. Parameter estimation experiments**
>
> **C5. Parameter estimation experiments.**
>
> We thank the reviewer for pointing out [R6, R7]. While these methods are designed for end-to-end parameter estimation in PDEs from a few measurements, they are not directly comparable to our experiments in Section 4.2. Our experiments focus on *self-supervised representation learning*, where we evaluate the quality of learned embeddings via linear probing. This is a more challenging setting than supervised parameter regression: the linear head is deliberately simple, intended as a stress test of the representations, rather than as a practical parameter prediction method. Linear probing is a standard evaluation in self-supervised representation learning, as seen in SimCLR [1], MoCo [2], MAE [3], DINO [4], etc., to name just a few, where it is used to assess the quality of learned embeddings independent of task-specific network heads.
>
> [1] Chen, T., Kornblith, S., Norouzi, M. and Hinton, G., 2020, November. A simple framework for contrastive learning of visual representations. In International conference on machine learning (pp. 1597-1607). PmLR.
>
> [2] He, K., Fan, H., Wu, Y., Xie, S. and Girshick, R., 2020. Momentum contrast for unsupervised visual representation learning. In Proceedings of the IEEE/CVF conference on computer vision and pattern recognition (pp. 9729-9738).
>
> [3] Caron, M., Touvron, H., Misra, I., Jégou, H., Mairal, J., Bojanowski, P. and Joulin, A., 2021. Emerging properties in self-supervised vision transformers. In Proceedings of the IEEE/CVF international conference on computer vision (pp. 9650-9660).
>
> [4] He, K., Chen, X., Xie, S., Li, Y., Dollár, P. and Girshick, R., 2022. Masked autoencoders are scalable vision learners. In Proceedings of the IEEE/CVF conference on computer vision and pattern recognition (pp. 16000-16009).

---

> ### Author Response · Authors · 2025-11-17
> **C6. Why is z (the global latent code) included?**
>
> **C6. Why is z (the global latent code) included?**
>
> The global latent code $z$ is included in the architecture primarily for *conditioning across different systems*, not for overfitting to a single instance. As discussed in our C1 and C5 responses, our framework emphasizes generalization: $z$ allows the network to capture higher-level system-specific information from a collection of trajectories, enabling predictions for unseen systems without tailoring the network to any one system.
>
> From an architectural perspective, $z$ functions analogously to the CLS token in Vision Transformers (ViT) [1], serving as a global query that aggregates information from all input trajectories and outputs the Hamiltonian value. A global token is now a widely adopted design in transformer architectures for predicting global properties or conditions, as it efficiently summarizes information from all input elements. This design allows the network to model global properties while keeping per-particle or per-step representations separate and reusable. Including $z$ in this way facilitates both structured representation learning and generalization across systems, consistent with the goals of our approach.
>
> [1] Dosovitskiy, A., Beyer, L., Kolesnikov, A., Weissenborn, D., Zhai, X., Unterthiner, T., Dehghani, M., Minderer, M., Heigold, G., Gelly, S., Uszkoreit, J. and Houlsby, N., 2021. An image is worth 16x16 words: Transformers for image recognition at scale. International Conference on Learning Representations (ICLR).

---

> ### Author Response · Authors · 2025-11-17
> **C7. Super-resolution experiment baseline**
>
> **C7. Super-resolution experiment baseline.**
>
> Learning-based super-resolution is meaningful when naive interpolation (linear, quadratic, etc.) is insufficient to recover the underlying information. In our experiments, we upsample from 16 timesteps to 128 timesteps, which is a sparse temporal signal. Furthermore, we aim to reconstruct physically meaningful trajectories, not just fill gaps numerically. This is analogous to image super-resolution, where learning-based methods recover semantic content beyond simple pixel interpolation (e.g., [1, 2]).
>
> We use CNNs as a baseline for super-resolution because our work focuses on *extending physical operators with guaranteed constraints to flexible tasks*, including trajectory upsampling. The CNN is unconstrained and serves as a non-physical baseline.
> Our approach demonstrates that physical operators (in our context, network operators that ensure certain physical constraints such as conservation laws) can be adapted to learning-based super-resolution while respecting conservation laws, whereas most prior works with guaranteed physics are limited to direct simulation tasks.
>
> [1] Ledig, C., Theis, L., Huszár, F., Caballero, J., Cunningham, A., Acosta, A., Aitken, A., Tejani, A., Totz, J., Wang, Z. and Shi, W., 2017. Photo-realistic single image super-resolution using a generative adversarial network. In Proceedings of the IEEE conference on computer vision and pattern recognition (pp. 4681-4690).
>
> [2]  Lim, B., Son, S., Kim, H., Nah, S. and Mu Lee, K., 2017. Enhanced deep residual networks for single image super-resolution. In Proceedings of the IEEE conference on computer vision and pattern recognition workshops (pp. 136-144).

---

> ### Author Response · Authors · 2025-11-17
> **C8. Usage of the term "neural operator"**
>
> **C8. Usage of the term "neural operator".**
>
> We appreciate the reviewer’s comment. The term “neural operator” introduced in [R5] (referring to mappings in function space) has been studied independently in multiple deep learning communities well before 2023. For example, the *functional map* framework and its follow-ups in computer vision and graphics [1–5] learn spectral bases, spectral features, and function-to-function mappings, and this is a widely recognized subarea in 3D computer vision and shape analysis. Notably, OperatorNet [5] (2019), four years prior to [R5], explicitly uses the term “functional operator”.
>
> More broadly, the concept of “operator” is longstanding in geometric deep learning and graph learning. The 2017 survey by Bronstein et al. [6] provides a comprehensive summary of spatial and spectral operators for non-Euclidean data – many of which remain central to modern architectures. Thus, while terminology has evolved across different communities, the underlying idea of operator-based neural mappings is not specific to a single line of work.
>
> Our intention in referencing this background is not to debate terminology or establish precedence, but simply to clarify that related concepts have emerged independently across fields, often with similar foundations but different motivations and implementations. In this work, our usage of “neural operator” reflects this broader perspective. As noted in our overall motivation, our framework draws inspiration from computer vision, which naturally leads to terminology and design choices that may differ from PDE-based neural operator literature. We hope this clarifies the intended meaning of the term within the context of our paper.
>
> [1] Litany, O., Remez, T., Rodola, E., Bronstein, A. and Bronstein, M., 2017. Deep functional maps: Structured prediction for dense shape correspondence. In Proceedings of the IEEE international conference on computer vision (pp. 5659-5667).
>
> [2] Attaiki, S., Li, L. and Ovsjanikov, M., 2023. Generalizable local feature pre-training for deformable shape analysis. In Proceedings of the IEEE/CVF Conference on Computer Vision and Pattern Recognition (pp. 13650-13661).
>
> [3] Cheng, X., Deng, C., Harley, A.W., Zhu, Y. and Guibas, L., 2024, September. Zero-shot image feature consensus with deep functional maps. In European Conference on Computer Vision (pp. 277-293).
>
> [4] Mitchel, T., Taylor, M.J. and Sitzmann, V., 2024. Neural isometries: Taming transformations for equivariant ml. Advances in Neural Information Processing Systems, 37, pp.7311-7338.
>
> [5] Huang, R., Rakotosaona, M.J., Achlioptas, P., Guibas, L.J. and Ovsjanikov, M., 2019. Operatornet: Recovering 3d shapes from difference operators. In Proceedings of the IEEE/CVF International Conference on Computer Vision (pp. 8588-8597).
>
> [6] Bronstein, M.M., Bruna, J., LeCun, Y., Szlam, A. and Vandergheynst, P., 2017. Geometric deep learning: going beyond euclidean data. IEEE Signal Processing Magazine, 34(4), pp.18-42.

---

> > ### Comment · Reviewer_SNKY · 2025-12-03
> > **Waiting for manuscript updation**
> >
> > I thank the authors for their detailed response. To fully understand their rebuttal, especially in the context of the paper's content, I must see the updated PDF with the changes promised in the rebuttal (preferably in a different color for easy viewing). If the new experiments are still running, please let us know.

---

> > > ### Author Response · Authors · 2025-12-03
> > >
> > > Thanks for the comment!
> > >
> > > In fact, we didn't realize that the discussion period also allows revisions to the PDF file -- we apologize for this, and will merge the explanations and the additional experiments into the file as soon as possible.
> > >
> > > The additional experiments are posted in our response to Reviewer kx7j. We will also merge them into the PDF file.

---

> > > > ### Comment · Action_Editor_pfMV · 2025-12-18
> > > >
> > > > Reviewer SNKY, now that the authors have uploaded their revised manuscript, please post your final recommendation (taking the revisions into account) as soon as possible. Thanks!

---

> ### Comment · Reviewer_SNKY · 2025-12-18
> **Reviewer SNKY's response to rebuttal**
>
> I thank the authors for their very detailed response and the updated PDF. Their response has addressed some of my concerns. I will submit my final recommendation after getting a couple of clarifications from the authors.
>
> **C1**. I thank the authors for providing a more detailed related works section. In my original comment, I had meant more recent works on Hamiltonian neural networks. Have there not been any relevant advances in Hamiltonian neural networks?
>
> **C2-6**. Your response addresses my concerns.
>
> **C7**. The authors replied that they "aim to reconstruct physically meaningful trajectories." Then, they say that "CNN...serves as a non-physical baseline." I agree with the first sentence, but not the second. I don't understand why CNN is a valid baseline, but a simple spline isn't. And why physics-informed baselines aren't.
>
> **C8**. I agree with the authors that the term "neural operators" is not limited to PINN literature. However, since the term is already in use in related literature, I think it's important to clarify the usage of this term. At the very least, the authors must include a short discussion in the paper justifying their use of the term.

---

### Review · Reviewer_kx7j · 2025-11-03

**Summary Of Contributions:**

This paper presents a physics-inspired reasoning framework for problem sets that enforce dynamical structures. The authors tackle issues stemming from current methodologies' obsession with local temporal prediction via simulation; instead, they propose a general-purpose neural operator with architectural innovations. The proposed denoising Hamiltonian network (DHN) is a unified model that extends Hamiltonian neural operators to capture non-local temporal dynamics, equipped with a denoising objective analogous to variational integration. The article demonstrates its performance on trajectory fitting for simulated data (pendulums) and relevant inference tasks on 1D or 2D degree-of-freedom (DOF) physical systems. The authors successfully demonstrate that the proposed architecture retains the scalability and robustness required for use in general physics-related tasks.

**Audience:**

Yes

**Audience Explanation:**

I find this work intriguing and believe it is novel in this field. Physics-based neural architectures that incorporate internal differential equation solvers inevitably struggle with the intrinsic numerical errors that arise during trajectory acquisition. The authors actively incorporated temporal architectures, such as transformers, which can effectively learn denoising spatiotemporal patterns within an erroneous learning environment. That said, the work would attract a broader audience if the proposed method demonstrated certain features, such as generality or scalability (as implied in the introduction), through more varied general experiments.

**Broader Impact Concerns:**

There is no concern.

**Claims And Evidence:**

Yes

**Claims Explanation:**

The paper is mostly sound. The key component of DHN is its use of discretized Hamiltonian dynamics with consecutive state blocks, which enables the capture of broader temporal correlations through a vectorized procedure. Another methodological improvement over traditional Hamiltonian neural networks is the masked modeling and denoising training scheme, which injects random perturbations into the input states (the “position” slots). By encouraging the model to correctly identify the original states, the authors claim this approach mitigates the cumulative errors that arise from numerical integration, a novel concept for PINN studies. Lastly, the authors seem to have incorporated a transformer-based decoding architecture to handle variable inputs and various masking schemes. I believe the experiments are standard and were appropriately conducted, but I did not go over every detail in the appendix.

**Requested Changes:**

* I think the introduction (Section 1) needs significant improvement to clarify the exact motivation and expected outcomes of the suggested improvements.
* The main argument seems to focus on the error-prone nature of local temporal prediction, an issue DHN claims to solve by examining global sequences via a masked training regime. However, the experiments show that leapfrog methods also demonstrate comparable improvements over forward Euler. This suggests that theoretical numerical analysis also has the potential to solve the outlined issue without the excessive memory and computational constraints of DHN. The authors are encouraged to disclose their computational costs in detail and discuss how theoretical work on Hamiltonian structures could benefit their work in the future.
* I believe the breadth of experiments in the current version is quite limited, even though the network architecture itself implies the universality of its solutions. The authors are encouraged to include other simulations besides pendulums or conduct more intensive ablation studies on the representation learning capabilities of the DHN framework. I also request the authors to test the DHN is scalable to multiple DOFs (>10), unless elaborate on why it is not scalable to multi-dimensional problems.

---

> ### Author Response · Authors · 2025-11-18
> **Additional experimental results: a 10-dimensional system**
>
> We thank the reviewer for the feedback and for suggesting additional experiments. Here, we include an experiment on a 10-dimensional physical system to evaluate the scalability and generalization ability of the proposed DHN framework.
>
> **Experimental setup.**
>
> We consider a spring chain with 10 particles, where a transverse wave travels from right to left. All particles share the same mass except for one “weird” particle whose index (position) and mass are randomly sampled. This creates a family of systems with varying local physical properties.
> The reason why we chose this system will be discussed at the bottom.
>
> The system looks like:
>
> o --- o --- o --- ...... --- O --- o --- o
>
> In the illustration, "o" means the standard particles, "O" means the "weird" particle, and "---" means the strings.
>
> We evaluate DHN on the forward simulation task, following the same training and test setup as Section 4.1 (“Completion on novel trajectories”).
>
> **Experimental results.**
>
> We report the average error on $q$ over time steps (in MSE$\downarrow$):
>
> | Time step                      |   15  |   31  |   47  |   63  |   79  |   95  |  111  |  127 |
> |--------------------------------|--------|-------|--------|--------|-------|-------|--------|--------|
> | ResNet (res blocks=1)  | 0.03 | 0.10 | 0.16 | 0.21 | 0.26 | 0.31 | 0.35 | 0.37 |
> | ResNet (res blocks=2)  | 0.05 | 0.18 | 0.29 | 0.37 | 0.43 | 0.47 | 0.48 | 0.48 |
> | Transformer                  | 0.03 | 0.29 | 1.00 | 2.06 | 3.34 | 4.77 | 6.31 | 7.96 |
> | HNN (forward Euler)     | 0.01 | 0.04 | 0.07 | 0.10 | 0.13 | 0.16 | 0.19 | 0.22 |
> | HNN (Leapfrog)            | 0.03 | 0.11 | 0.19 | 0.27 | 0.33 | 0.39 | 0.44 | 0.49 |
> | Ours (block size=2)      | 0.05 | 0.20 | 0.34 | 0.46 | 0.56 | 0.65 | 0.74 | 0.82 |
> | Ours (block size=4)      | 0.01 | 0.05 | 0.08 | 0.10 | 0.13 | 0.15 | 0.17 | 0.19 |
> | Ours (block size=8)      | 0.01 | 0.05 | 0.09 | 0.12 | 0.14 | 0.17 | 0.19 | 0.21 |
>
> Observations:
> - DHN with larger block sizes (4 or 8) achieves the best results, particularly at longer time horizons.
> Larger block sizes reduce the number of autoregressive rollouts (because multiple steps are predicted in parallel), thus mitigating temporal error accumulation.
> - Interestingly, HNN with Leapfrog performs even worse than with forward Euler, despite Leapfrog being a more accurate integrator.
> Since the test set contains unseen physical systems, this suggests that the performance bottleneck is not the numerical integrator but rather the model’s ability to fit and generalize across systems.
>
> **Discussions, limitations, and future work.**
>
> Our transformer architecture focuses on modeling **temporal relations**, while its spatial processing relies only on a lightweight MLP. Theoretically, the model can handle systems of arbitrary dimensionality; however, the practical limitation lies in how well the network can capture **spatial structure**, not simply in the number of degrees of freedom.
>
> We chose the 10-particle spring chain deliberately because its spatial structure is simple and nearly linear, making it suitable for an MLP. For more complex multi-particle or high-dimensional systems, incorporating stronger spatial inductive biases would likely improve performance. Potential extensions include:
> - PointNet-style modules for permutation-invariant particle sets;
> - Graph neural networks for arbitrary connectivity;
> - Full spatiotemporal transformers combining spatial attention with temporal attention.
>
> We believe these are promising directions for scaling DHN to more general physical systems.

---

> ### Author Response · Authors · 2025-11-18
> **Answers to other questions**
>
> **Clearer explanation of motivation.**
>
> We thank the reviewer for the helpful suggestion regarding the introduction. Our motivation is to draw inspiration from other deep learning domains (mostly computer vision) to study physical learning from new perspectives, both in terms of problem formulation and model design. This makes the framing less like a ‘typical’ physical learning paper, and we agree that the introduction should make this motivation more explicit. Broadly, existing approaches fall into two extremes: (i) physics-driven operators that enforce structure by construction (e.g., equivariant models, Hamiltonian networks), which offer guarantees but limited scalability; and (ii) highly expressive black-box neural networks, which scale well but tend to capture semantic rather than physical structure. Our goal is to explore the space between these extremes by designing neural operators that retain key physical constraints while preserving the flexibility of modern deep networks. We will revise the introduction to articulate this motivation more clearly.
>
> We will also clarify the expected gains of our approach. Compared to strictly physics-based models, our framework provides:
> - Greater task flexibility, allowing the same framework to support diverse tasks -- not only forward simulation but also the downstream applications in Sections 4.1–4.3. These experiments are included primarily to demonstrate the range of capabilities enabled by our formulation, rather than to emphasize performance alone.
> - Stronger latent representations that encode higher-level physical information (e.g., Section 4.2), analogous to the progression from low-level to high-level features in computer vision.
> - Improved performance within each task due to model components that balance physical constraints with the versatility of black-box learning.
>
> We will integrate these points into the revised introduction to make the motivation and expected outcomes clearer.
>
> **Comparison to theoretical numerical analysis.**
>
> We thank the reviewer for raising this important point about numerical accuracy versus learning-based accuracy. We agree that clearer explanations and discussions on this topic would strengthen the paper, and we will include additional clarifications in our revised version.
>
> *(I) Motivation for denoising.*
>
> The denoising mechanism in DHN is primarily designed to improve **data efficiency**, instead of computational efficiency. Its core purpose is to allow the model to perform iterative refinement of predicted future states, analogous to the optimization performed in variational integrators, which achieve high long-term accuracy by evaluating the Hamiltonian both on and off the true trajectory.
>
> However, in a learning setup:
> - We only observe states on the trajectory (“positive data”).
> - We do **not** have access to off-trajectory states (“negative data”), which would be required to shape the energy landscape as variational integrators do.
> - Obtaining such off-manifold information would dramatically increase data requirements.
>
> This creates a fundamental difficulty: how to learn accurate next-state predictions without supervision on the surrounding energy landscape. In numerical computation, one can perform forward integration using only the trajectory, but higher-accuracy methods (such as variational integrators) depend on energy values at both on-trajectory and off-trajectory points. This gap underscores why the learning problem is more challenging than classical numerical integration.
>
> Here, we draw inspiration from diffusion models, which face an analogous challenge in generative modeling. They construct a “handcrafted” energy in the free space around true data so that the model can iteratively refine predictions toward the data manifold. DHN adopts this philosophy:
> - It learns the true energy on-trajectory,
> - And a *learned surrogate* energy off-trajectory,
> - Sufficient to ensure that predicted states are progressively pulled toward the correct trajectory.
>
> This is the idea we wanted to illustrate in Figure 3 left.
>
> *(II) Why better integrators doesn't solve the problem?*
>
> Our new experiment on the 10D system also shows that Leapfrog cannot help -- it even performs worse than Euler on unseen systems.
> This happens because:
> - On the test set, the model must generalize to new physical parameters.
> - The dominant source of error is therefore model misgeneralization, not the accuracy of the numerical integrator.
> - A better integrator cannot compensate for errors in the learned dynamics or the learned Hamiltonian.
>
> Thus, the limitation arises from **learning**, not from numerical computation.

---

### Author Response · Authors · 2025-12-17
**Revision summary**

This work aims to bring together intuitions from different domains in deep learning, combining expressivity (e.g., high-level “semantic-like” features in representation learning) with physical interpretability in scientific deep learning. The initial draft was primarily written from a computer vision and representation learning perspective. For example, while we introduced Hamiltonian networks as preliminaries, we included relatively little discussion of transformer architectures, such as global tokens, decoder-only transformers, and auto-decoders, which are widely used in domains like computer vision and NLP.

In the revision, we add more background discussion, related work, and design explanations, as we expect the readership of this paper to come from diverse research communities. Since our work draws motivation from multiple areas, it is difficult to exhaustively cover all relevant literature and technical details from every domain. We therefore focus on discussions that are most directly aligned with our core motivations and contributions.

Below is a summary of the revisions:
- Section 1: Explaining the motivations for the experiments, and how each experimental result relates to the properties of our network design.
- Section 2: Expanded related work discussions.
    - An additional paragraph on “Symmetry-constrained neural networks,” discussing how our method relates to enforcing and relaxing symmetry in neural networks from a broader perspective (as conservation laws in physical systems are also a form of symmetry).
    - More explanations of the general area of “Physics-informed and operator-based methods.”
- Section 3.4: References for the global token design in commonly used transformer architectures, with a brief explanation.
- Section 4 (beginning): A concise overview of the experiments and expected gains, to help readers interpret the results.
- Section 4.1: Additional experiments on higher-dimensional systems (traveling waves on spring chains).
- Section 4.3: An explanation of learning-based super-resolution, with references to analogous tasks in computer vision.

The revised sentences and paragraphs are marked in blue to distinguish them from the original text. For better readability, we focus here on the major revisions, while leaving minor edits (e.g., figure formatting and small wording changes) to future revisions.

---

### Author Response · Authors · 2025-12-31
**question about the discussion process**

Dear Action Editor,

We may feel perplexed: as the latest questions were raised by the reviewer far after the discussion period ended, should we reply to the question and participate in the discussion, or would that be against the submission guidelines? We provided very detailed responses to all reviewers' initial questions and comments, but didn't see any questions during the discussion period. Thus the questions asked 17 days after the discussion period ended were a bit unexpected to us -- was that because we missed some questions during the discussion period, e.g., due to system errors? Would be very grateful if you could offer more instructions. Great thanks! (Also apologize for sending this message during the holiday, if it bothers you.)

---

> ### Comment · Action_Editor_pfMV · 2026-01-07
>
> Hello authors,
>
> Sorry I didn't get to your question earlier; I took a break for the holiday season.
>
> Because TMLR is a journal with rolling submissions, we have more flexibility in the timeline than is typical for a conference review process. In this case, the reviewers can perhaps be forgiven for making late requests for clarification as the author response was submitted at the very end of the discussion period. In general I'd prefer that reviewers have as much information as they need to make fair, informed decisions than to stick strictly to the deadlines so I've been permissive of allowing follow-up discussion after the discussion period (including the request for a revised manuscript).
>
> Nevertheless, at this point the particular question seems to be moot as the reviewer ultimately felt able to make a recommendation without the answers to their most recent questions. With all recommendations submitted, we'll now move into the decision phase. Thanks all for your engagement regarding this paper!

---

### Decision · Action_Editor_pfMV · 2026-01-14

**Recommendation:** Accept as is

**Audience:**

Yes

**Audience Explanation:**

All reviewers agree that the paper addresses important problems in interesting ways and that the findings are likely to be relevant to the interests of the TMLR readership. To improve on this even more, the authors are encouraged to consider the reviewers' final advice/feedback when developing the camera-ready version. Specfically:

Section 1. The authors have specifically clarified the motivation behind their experiments. While this improves clarity, I had anticipated a more comprehensive motivation regarding the proposed framework itself. If possible, I encourage the authors to strengthen the introduction by grounding the framework’s theoretical or practical necessity more thoroughly.

C1. I thank the authors for providing a more detailed related works section. In my original comment, I had meant more recent works on Hamiltonian neural networks. Have there not been any relevant advances in Hamiltonian neural networks?

**Claims And Evidence:**

Yes

**Claims Explanation:**

By and large the reviewers agree that the paper supports its claims well. One reviewer felt that more empirical work was needed to explore the behavior of the proposed approach in higher dimensional problems (even after the addition of new results in the revised manuscript). From my own read of the paper, I can see that the reviewers suggestions could strengthen the paper but did not find that the paper significantly overclaimed relative to the available evidence. The authors are encouraged to consider the reviewers' final advice/feedback when developing the camera-ready version. Specifically:

New experiment. I again thank the authors for their effort in providing new empirical results. Regarding my previous query, they provided a counterexample where HNN (Euler) outperforms HNN (LeapFrog). It would be valuable to investigate whether the Euler method exhibits a consistent tendency to be robust as dimensionality increases. More importantly, my interpretation of the new figure (Fig. 8) is that HNN (Euler) significantly outperforms one of the DHN settings (block size=2) and performs comparably to the best-performing setting (block size=4/8) in this single trial. The rightmost subfigure in Fig. 8 also makes it difficult to discern the particular strength of HNN in the 10-dimensional setting. I understand DHN outperforms other baselines in the table format; I suggest the authors refine their model's presentation in complex scenarios, especially given that HNN offers greater simplicity for practical deployment.

C7. The authors replied that they "aim to reconstruct physically meaningful trajectories." Then, they say that "CNN...serves as a non-physical baseline." I agree with the first sentence, but not the second. I don't understand why CNN is a valid baseline, but a simple spline isn't. And why physics-informed baselines aren't.

C8. I agree with the authors that the term "neural operators" is not limited to PINN literature. However, since the term is already in use in related literature, I think it's important to clarify the usage of this term. At the very least, the authors must include a short discussion in the paper justifying their use of the term.